# Resolving the nonequilibrium Kondo singlet in energy- and position-space using quantum measurements

Andre Erpenbeck[1,2] and Guy Cohen[1,2*]

**1** The Raymond and Beverley Sackler Center for Computational Molecular and Materials Science, Tel Aviv University, Tel Aviv 6997801, Israel
**2** School of Chemistry, Tel Aviv University, Tel Aviv 6997801, Israel

★ gcohen@tauex.tau.ac.il

## Abstract

The Kondo effect, a hallmark of strong correlation physics, is characterized by the formation of an extended cloud of singlet states around magnetic impurities at low temperatures. While many implications of the Kondo cloud's existence have been verified, the existence of the singlet cloud itself has not been directly demonstrated. We suggest a route for such a demonstration by considering an observable that has no classical analog, but is still experimentally measurable: "singlet weights", or projections onto particular entangled two-particle states. Using approximate theoretical arguments, we show that it is possible to construct highly specific energy- and position-resolved probes of Kondo correlations. Furthermore, we consider a quantum transport setup that can be driven away from equilibrium by a bias voltage. There, we show that singlet weights are enhanced by voltage even as the Kondo effect is weakened by it. This exposes a patently nonequilibrium mechanism for the generation of Kondo-like entanglement that is inherently different from its equilibrium counterpart.



# 1   Introduction

Strongly correlated quantum systems are a central paradigm in condensed matter physics. A pivotal role in this field is played by the Kondo effect [1, 2], where the resistance of metals with a small concentration of magnetic impurities increases at low temperatures. This is due to electrons within the impurities becoming intricately entangled with those in the surrounding bulk material [3]. The resulting low energy state is characterized by a narrow resonance in the spectral function [4], and by singlet correlations that extend far beyond the impurity [5]. The latter are believed to cause enhanced scattering in a volume that may be orders of magnitude larger than that of the impurity atom [2–6].

The correlated singlet is known in the literature as the Kondo screening cloud, and its equilibrium properties are well understood in a wide variety of circumstances. The length scale characterizing this cloud can be estimated from scaling or perturbative arguments [7, 8] and explicitly calculated numerically [9, 10]. Predictions can then be made about the experimentally observable implications of the existence of the Kondo cloud [11]. Important examples include oscillations in density and spin correlations [10, 12–19]; dependence on finite size effects or boundary conditions in the metallic environment [20–26]; and entanglement between the dot and conduction electrons [27, 28]. The dynamical formation of equilibrium density oscillations and spin correlations after a quantum quench has also been explored [29, 30].

Experimental studies have confirmed many of the predicted microscopic consequences of the existence of the Kondo cloud, beyond its macroscopic effect on conductance. To give a few examples, the cloud's effect on electronic spin polarizability could in principle be measured by nuclear magnetic resonance (NMR) experiments, though this is difficult [31]. Size dependent effects in nanoscale systems were detected [32–39]. Perhaps the most direct observations come from studies combining scanning tunneling microscopy and spectroscopy [40], which have generated evidence that electrons scatter off the cloud [41].

Some of the clearest and most controlled spectroscopic observations of the Kondo effect [42, 43], as well as demonstrations of the size of the associated cloud [44], are obtained in mesoscopic transport experiments. Here, the impurity embedded in a metallic host is replaced by a quantum dot spanning two noninteracting leads. Within linear response, the conductance across this junction provides access to the spectral function of the dot; and can also probe its nonequilibrium properties. An important example is the prediction that the Kondo resonance can be split by a bias voltage before being destroyed by nonequilibrium dissipation [45–49]. A different resonance then resides (approximately, see Ref. [49]) at the chemical potential of each lead. However, it remains unknown to what degree these split resonances correspond to the equilibrium Kondo resonance and whether they share its singlet-like nature. It is also largely unknown whether nonequilibrium currents are capable of suppressing, enhancing or distorting the Kondo cloud.

Despite all this progress, the Kondo cloud itself—in the sense of an extended singlet—has yet to be directly observed in either equilibrium or nonequilibrium situations. Even though the extended singlet is arguably the defining quality of the Kondo cloud, there has been virtually

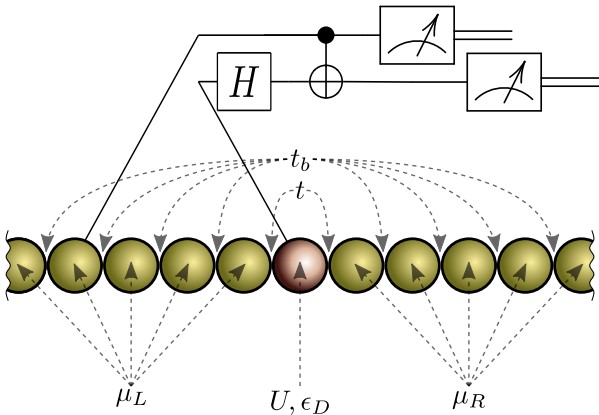

Figure 1: Schematic representation of the system under investigation. The quantum impurity (bronze circle) is coupled to semi-infinite chains of identical atoms (gold circles). A simultaneous quantum measurements on the impurity and on the chains can quantify the Kondo phenomenon.

no direct study of its structure in either theory or experiment. This is understandable, because the degree to which a system exhibits singlet correlations is difficult to measure compared with the observables on which most work has been focused. It is nevertheless important to realize that while the existence of a Kondo singlet implies, e.g., oscillatory response in spin–spin correlations [9], the converse is not necessarily true.

On the other hand, singlet correlations not related to Kondo physics have been experimentally measured in several types of very different experimental protocols. For example, in optics experiments, knowledge about singlets between entangled pairs of photons can be extracted [50–52]. Furthermore, in NMR experiments singlets between nuclear spins can be observed by way of specialized pulse sequences [53, 54]. As a third example, in ultracold atomic systems, singlet and triplet states can be artificially manufactured and controlled [55].

From the quantum information point of view, measuring the projection on a singlet state could be considered a specialized kind of "quantum measurement". It requires a transformation from the Bell (i.e. singlet–triplet) basis to a so-called computational basis, where measurements of normal correlation functions are carried out. This is accomplished by a simple quantum circuit (an inverse Bell circuit, see top part of Fig. 1), which may be implemented in different ways within different experiments. Therefore, in a system enabling implementation of generic two-qubit quantum gates–e.g., as was recently suggested for ultracold fermionic gases [56]–a singlet projection measurement would be relatively straightforward. Quantum tomography is another potentially viable route to accessing such quantum observables in correlated electron systems [57].

Experiments of this sort on Kondo systems have yet to be performed, and clearly represent a significant technical challenge. Nevertheless, it is important to distinguish between observables that are theoretically interesting, but not generally measurable; and observables that may be difficult to access in experiment, but are measurable in principle. Bipartite entanglement entropy is one example of an observable that is often discussed in the literature [27], but generally belongs to the first class. The projection onto a (two-particle) singlet state, our main focus in the rest of this manuscript, is of the latter variety.

In the following, we present a study of singlet correlations in the nonequilibrium (and equilibrium) Anderson impurity model, where the impurity is modeled by a single, spin degenerate electronic orbital. Complementary representations of singlet correlations in energy- and position-space are considered, allowing us to construct a detailed picture of the Kondo

cloud in several regimes. In particular, we establish that singlet correlations are an excellent and intuitive observable for examining the well-understood equilibrium physics of the Kondo cloud. Then, we show that they can provide new insight about the nonequilibrium physics.

To solve the nonequilibrium impurity problem, we use the propagator flavor of the noncrossing approximation (NCA) [58–60]. Since its introduction to the field [61–64], variants of the NCA and its extensions have been used to study various aspects of the nonequilibrium Kondo effect [45, 46, 58, 59, 65–67]. The method provides qualitatively, though not quantitatively, accurate results at higher temperatures in the Kondo regime, and its regions of applicability have often been explored [59, 60, 67–70]. However, it cannot be used to systematically examine, e.g., the scaling limit that emerges at low energies, where at least vertex corrections are needed [71–74] and numerically exact methods are desirable. The NCA used here is the lowest order precursor of the numerically exact bold-line Monte Carlo [48, 60, 68, 75, 76] and Inchworm Monte Carlo [49, 77–84] methods. Other recent approaches to the impurity problem may also be applicable to the same problem [85–89], and revisiting this work within a controlled numerical scheme will be a goal for future studies.

The outline of the paper is as follows: In Sec. 2, we introduce the model and provide general definitions of singlet observables. Sec. 3 is dedicated to the NCA method and its application to such observables. Our results, first in equilibrium and then with a nonequilibrium bias, are presented in Sec. 4. Finally, in Sec. 5, we discuss our conclusions.

# 2 Hamiltonian and observables

## 2.1 Anderson impurity model

We consider the Anderson impurity model, which is often used to describe a quantum dot with electron–electron interactions coupled to noninteracting leads. The system is described by the Hamiltonian

$$H = H_D + H_B + H_{DB}. \tag{1}$$

Here, the internal dot Hamiltonian $H_D$, in units where $\hbar = e = 1$, is

$$H_D = \sum_\sigma \epsilon_D d_\sigma^\dagger d_\sigma + U d_\uparrow^\dagger d_\uparrow d_\downarrow^\dagger d_\downarrow, \tag{2}$$

where the $d_\sigma^{(\dagger)}$ annihilate(create) a spin $\sigma \in \{\uparrow, \downarrow\}$ electron on the dot. $\epsilon_D$ is the single-particle occupation energy, and $U$ is the energetic cost of Coulomb charging when the dot is doubly occupied. The lead Hamiltonian $H_B$ represents a continuum of noninteracting electrons,

$$H_B = \sum_\ell \sum_{k \in \ell} \sum_\sigma \epsilon_k a_{k\sigma}^\dagger a_{k\sigma}. \tag{3}$$

The lead index $\ell \in \{L, R\}$ stands for the "left" and "right" lead, respectively. The $a_{k\sigma}^{(\dagger)}$ annihilate(create) an electron with spin $\sigma$ and energy $\epsilon_k$ on lead orbital $k$ in either lead. Finally, the dot–lead coupling is

$$H_{DB} = \sum_\ell \sum_{k \in \ell} \sum_\sigma \left( V_k a_{k\sigma}^\dagger d_\sigma + \text{h.c.} \right). \tag{4}$$

The coupling constants $V_k$ are determined by the lead coupling density

$$\Gamma_\ell(\epsilon) = \pi \sum_{k \in \ell} |V_k|^2 \delta(\epsilon - \epsilon_k). \tag{5}$$

## 2.2 Singlet weights and projectors

Our next task is to construct a set of observables that directly relate to Kondo correlations between the dot and specific bath orbitals. We will do this in two steps. First, we will consider projections onto specific dot–bath singlet-states and construct second-quantized operators associated with them. Then, we will argue that these operators still contain some non-Kondo contributions, and discuss how they can be removed.

Let $\chi$ be an index characterizing a lead orbital. The exact meaning of $\chi$ will not yet be further specified, so that it can denote either a single particle eigenfunction of the lead Hamiltonian or a local lead orbital. In general, however, $a_{\chi\sigma}^{(\dagger)}$ is a linear combination of the $a_{k\sigma}^{(\dagger)}$. These operators generate a local subspace on orbital $\chi$ that contains the zero electron state $|0_\chi\rangle$, the one electron states $|\uparrow_\chi\rangle$ and $|\downarrow_\chi\rangle$ and the two electron state $|\uparrow\downarrow_\chi\rangle$. The dot operators $d_\sigma^{(\dagger)}$ similarly generate the states $|0_D\rangle$, $|\uparrow_D\rangle$, $|\downarrow_D\rangle$ and $|\uparrow\downarrow_D\rangle$.

Consider, then, a two particle singlet-state formed between the dot orbital $D$ and the lead orbital $\chi$. The wavefunction of this state can be written as follows:

$$|s_\chi\rangle = \frac{1}{\sqrt{2}}\Big(|\uparrow_D\downarrow_\chi\rangle - |\downarrow_D\uparrow_\chi\rangle\Big). \tag{6}$$

If we define the operators

$$\begin{aligned}
P_{\chi\chi'\chi''\chi'''}^{\sigma\sigma'} &\equiv \Big(a_{\chi\sigma}^\dagger a_{\chi'\sigma} d_{\sigma'}^\dagger d_{\sigma'}\Big)\cdot\Big(a_{\chi''\sigma'} a_{\chi'''\sigma'}^\dagger d_\sigma d_\sigma^\dagger\Big),\\
E_{\chi\chi'}^{\sigma\sigma'} &\equiv a_{\chi\sigma}^\dagger a_{\chi'\sigma'} d_{\sigma'}^\dagger d_\sigma,
\end{aligned} \tag{7}$$

the projector onto $|s_\chi\rangle$ can be expressed as

$$|s_\chi\rangle\langle s_\chi| = \frac{1}{2}\Big(P_{\chi\chi\chi\chi}^{\uparrow\downarrow} + P_{\chi\chi\chi\chi}^{\downarrow\uparrow} - E_{\chi\chi}^{\uparrow\downarrow} - E_{\chi\chi}^{\downarrow\uparrow}\Big). \tag{8}$$

Here, $P_{\chi\chi\chi\chi}^{\sigma\sigma'}$ selects the state $|\sigma'_D\sigma_\chi\rangle$, and $E_{\chi\chi}^{\sigma\sigma'}$ exchanges a spin between the dot orbital and the lead orbital $\chi$. We remark in passing that it is similarly possible to express projectors onto other states, such as the triplet-states

$$\begin{aligned}
|t_{1\chi}\rangle &= |\uparrow_D\uparrow_\chi\rangle\,,\\
|t_{2\chi}\rangle &= \frac{1}{\sqrt{2}}\Big(|\uparrow_D\downarrow_\chi\rangle + |\downarrow_D\uparrow_\chi\rangle\Big),\\
|t_{3\chi}\rangle &= |\downarrow_D\downarrow_\chi\rangle\,,
\end{aligned} \tag{9}$$

in terms of $P_{\chi\chi\chi\chi}^{\sigma\sigma'}$ and $E_{\chi\chi}^{\sigma\sigma'}$. This enables the application of our methodology to a variety of physical questions beyond those to be considered here. Analogous expressions for multi-orbital impurities can be devised accordingly.

The operator $|s_\chi\rangle\langle s_\chi|$ was designed specifically to extract singlet correlations, but still admits contributions that might be considered trivial. For example, $P_{\chi\chi\chi'\chi'}^{\sigma\sigma'}$ does not eliminate the product state $|\sigma'_D\sigma_\chi\rangle$, which can occur even in a system where the dot and leads are neither coupled nor entangled. While this state is characterized by ("classical", or population-based) spin–spin correlations, it is not necessarily indicative of quantum correlations, and we discard it in the remainder of this work. To do this in practice, wherever $\langle P_{\chi\chi'\chi''\chi'''}^{\sigma\sigma'}\rangle$ might appear we replace it with the quantity

$$\langle P_{\chi\chi'\chi''\chi'''}^{\sigma\sigma'}\rangle_{\text{correl}} \equiv \langle P_{\chi\chi'\chi''\chi'''}^{\sigma\sigma'}\rangle - \delta_{\chi\chi'}\delta_{\chi''\chi'''}\langle d_{\sigma'}^\dagger d_{\sigma'} d_\sigma d_\sigma^\dagger\rangle f_\chi \bar{f}_{\chi''}. \tag{10}$$

Here, $f_\chi$ is the Fermi function (or initial occupation probability) associated with lead orbital $\chi$, and $\bar{f}_\chi = 1 - f_\chi$. It is important to note that for the sake of simplicity, this definition neglects nonequilibrium corrections to the lead occupancy $\langle a_{\chi\sigma}^\dagger a_{\chi\sigma}\rangle$.

The expectation value of the correlated singlet weight operator is given by

$$s(\chi) \equiv \frac{1}{2}\Big( \langle P^{\uparrow\downarrow}_{\chi\chi\chi\chi} \rangle_{\text{correl}} + \langle P^{\downarrow\uparrow}_{\chi\chi\chi\chi} \rangle_{\text{correl}} - \langle E^{\uparrow\downarrow}_{\chi\chi} \rangle - \langle E^{\downarrow\uparrow}_{\chi\chi} \rangle \Big), \tag{11}$$

whereby we emphasize that all operators are evaluated at the same time. To simplify the notation, the "correl" subscript will be dropped from now on where no confusion can occur. The significance of $s(\chi)$ is self evident in light of the singlet nature of Kondo physics, and will be demonstrated with several examples in Sec. 4.

## 3 Methodology

The singlet weight $s(\chi)$ is a well-defined quantity, and in principle a variety of numerically exact methods could be adapted to evaluating the corresponding expectation values in a controlled manner. However, in the present context we plan to explore general qualitative aspects, such that an approximate treatment suffices. In this section, we discuss the approximation scheme that will be used to evaluate $s(\chi)$ in the present work, the noncrossing approximation (NCA). Sec. 3.1 explains how the operators $P^{\sigma\sigma'}_{\chi\chi'\chi''\chi'''}$ and $E^{\sigma\sigma'}_{\chi\chi'}$ are treated and how $s(\chi)$ is obtained for a general orbital index $\chi$. Given this, Sec. 3.3 specializes the discussion to the energy representation $\chi = k$, while Sec. 3.4 specializes it to the position representation $\chi = x$.

### 3.1 Noncrossing approximation and the vertex function

Our treatment of the model will be based on the NCA, a self-consistent, lowest order perturbative expansion in the dot–lead coupling/hybridization [46, 58–60, 65–67, 69, 76, 90–92]. Generally, the name NCA refers to a class of hybridization expansions which only account for contributions that have a diagrammatic representation in which the hybridization lines do not cross. NCA methods are rooted in the seminal work by Grewe and Kuramoto [61, 62], which forms the basis for various extensions that account for finite electron–electron interaction strengths [71, 93] and nonequilibrium conditions [46]. In its basic formulation, the NCA successfully captures the physics at temperatures that are not far below the Kondo temperature, as well as in the large $U$ limit and for small bias voltages. However, it does not correctly reproduce the Kondo behavior in the scaling regime. For this regime, vertex corrections have proven to be essential [59, 72–74]. These extensions of the NCA, which are also numerically more demanding, have been successful in recovering the temperature scaling behavior characterizing Kondo phenomena in agreement with numerical renormalization group calculations [94, 95].

Here, we provide a brief overview of the method focusing on the details needed to discuss the evaluation of singlet weights in the next subsection. For a more systematic introduction to the propagator NCA, we refer the reader to the literature [60].

The expectation value of a dot operator $A$ at time $t$ is given by:

$$\langle A(t) \rangle = \text{Tr}\big(\rho U^{\dagger}(t) A U(t)\big). \tag{12}$$

Here, $\rho = \rho_D \otimes \rho_B$ is the initial density matrix, which we assume to be a product of an initial dot state $\rho_D$ and an initial lead state $\rho_B$; and $U(t) = \text{T}\exp(-i\int_0^t H(\tau)d\tau)$ is the time evolution operator, with T the time ordering operator. Let us define the vertex function,

$$K^{\beta}_{\alpha}(t, t') = \text{Tr}_B\big\{\rho_B \langle \alpha| U^{\dagger}(t)|\beta\rangle \langle\beta| U(t')|\alpha\rangle\big\}, \tag{13}$$

such that the expectation value from Eq. (12) can be expressed as

$$\langle A(t) \rangle = \sum_{\beta} K^{\beta}_{\alpha}(t, t) \langle\beta|A|\beta\rangle. \tag{14}$$

(a)

(b)

(c)

Figure 2: (a) Diagrammatic representation of the Dyson equation for the vertex function, Eq. (15). (b) Diagrammatic representation of the Dyson equation for the propagator, Eq. (20). (c) Examples of contributions to $\langle P^{\sigma\sigma'}_{\chi\chi'\chi''\chi'''}\rangle$ and $\langle E^{\sigma\sigma'}_{\chi\chi'}\rangle$ in Eqs. (24) and (25). The curled ("gluon") lines denote bath correlation functions connected to the observable at the measurement time $t$.

Here, the $\alpha$ and $\beta$ indices enumerate a basis of many-particle states in the dot subspace, and we assume that the initial state of the isolated dot can be written in the form $\rho_D = |\alpha\rangle\langle\alpha|$.

The hybridization expansion finds $K^\beta_\alpha(t, t')$ by perturbatively expanding in the dot–lead coupling $H_{DB}$. As such, $K^\beta_\alpha(t, t')$ is given by the Dyson equation

$$K^\beta_\alpha(t, t') = k^\beta_\alpha(t, t') + \sum_{\gamma\delta}\int_0^t\int_0^{t'} d\tau_1 d\tau'_1\, k^\beta_\delta(t - \tau_1, t' - \tau'_1)\, \xi^\delta_\gamma(\tau_1 - \tau'_1)\, K^\gamma_\alpha(\tau_1, \tau'_1). \tag{15}$$

A diagrammatic representation of this equation is shown in Fig. 2(a). The quantity $k^\beta_\alpha(t, t')$ will be defined later. Within the NCA, the exact cross-branch self-energy $\xi^\beta_\alpha(t)$ is replaced with an approximate form that only takes into account the lowest nonvanishing order in the expansion,

$$\xi^\beta_\alpha(t) \simeq \sum_{\sigma,\ell\in\{L,R\}}\left(\Delta^<_\ell(t)\,\langle\alpha|d_\sigma|\beta\rangle\,\langle\beta|d^\dagger_\sigma|\alpha\rangle + \Delta^>_\ell(t)\,\langle\alpha|d^\dagger_\sigma|\beta\rangle\,\langle\beta|d_\sigma|\alpha\rangle\right). \tag{16}$$

Here, the lesser and greater hybridization functions, $\Delta^<_\ell(t)$ and $\Delta^>_\ell(t)$, are determined by the lead coupling density $\Gamma_\ell(\epsilon)$ and the initial equilibrium distributions in the leads, $f_\ell(\epsilon)$:

$$\Delta^<_\ell(t) = \frac{1}{\pi}\int d\epsilon\, e^{+i\epsilon t}\, \Gamma_\ell(\epsilon)f_\ell(\epsilon), \tag{17}$$

$$\Delta^>_\ell(t) = \frac{1}{\pi}\int d\epsilon\, e^{-i\epsilon t}\, \Gamma_\ell(\epsilon)\bar{f}_\ell(\epsilon). \tag{18}$$

When the Dyson equation is solved self-consistently, the NCA effectively incorporates an infinite subset of all possible perturbative contributions. In this context, the name NCA refers to the fact that the methodology only includes contributions with a diagrammatic representation where the hybridization lines do not cross [60].

We now return to the remaining undefined quantity in Eq. (15),

$$k^\beta_\alpha(t, t') \equiv \delta_{\alpha\beta} G^*_\beta(t) G_\alpha(t'), \tag{19}$$

which contains all contributions to the vertex function with hybridization events limited to a single branch of the Keldysh contour, and which is given in terms of the single-branch propagator $G_\alpha(t) = \langle \alpha | \mathrm{Tr}_B (\rho_B U(t)) | \alpha \rangle$. Note that we have taken $G_\alpha(t)$ to be diagonal in the dot state basis, which is possible for the model used here but not in general. Like the vertex function, the propagator can be written as the solution of a Dyson equation,

$$G_\alpha(t) = g_\alpha(t) - \int_0^t \int_0^{\tau_1} d\tau_1 d\tau_2 \, g_\alpha(t - \tau_1) \Sigma_\alpha(\tau_1 - \tau_2) G_\alpha(\tau_2), \tag{20}$$

a diagrammatic representation of which is provided in Fig. 2(b). In the NCA, we consider only the lowest-order contribution to the single-branch self-energy:

$$\Sigma_\alpha(t) \simeq \sum_{\beta,\sigma,\ell \in \{L,R\}} \left( \Delta_\ell^<(t) \cdot \langle \alpha | d_\sigma | \beta \rangle \langle \beta | d_\sigma^\dagger | \alpha \rangle + \Delta_\ell^>(t) \cdot \langle \alpha | d_\sigma^\dagger | \beta \rangle \langle \beta | d_\sigma | \alpha \rangle \right) G_\beta(t). \tag{21}$$

Finally, $g_\alpha(t) = e^{-iE_\alpha t}$ is the atomic propagator obtained in the absence of a coupling between the dot and the leads. This can be obtained directly from the state energies $E_\alpha$ of the isolated dot, which can in turn be found analytically in the present model.

To this point, we have described how to calculate the expectation value of an observable as it evolves in time $t$ from the moment where the dot and leads are connected. It is also possible, and in fact substantially easier, to directly calculate steady state expectation values using the NCA framework. This is because at steady state, the vertex function depends only on the difference between its two time arguments:

$$K_\alpha^\beta \left( t, t' \right) \xrightarrow[t,t' \to \infty]{} K^\beta \left( t - t' \right). \tag{22}$$

This is equivalent to requiring that any dependence on the initial condition has faded away with time, and that time-local observables have become independent of time. By definition, the NCA propagator $G_\alpha(t)$ already depends only on a single time argument. Consequently, the vertex function at steady state must obey

$$K^\beta(t) = \int_{-\infty}^t d\tau_1 \int_{\tau_1}^\infty d\Delta\tau \, G_\beta^*(t - \tau_1) G_\beta(\Delta\tau - \tau_1) \sum_\gamma \xi_\gamma^\beta(\Delta\tau) K^\gamma(\Delta\tau), \tag{23}$$

which directly follows from Eq. (15) upon neglecting the initial condition and propagating from the infinite past. Eq. (23) is therefore iterated until self-consistency is established. However, Eq. (23) has no inhomogeneous initial condition term, and is linear in the vertex function. Its solutions are therefore unbound with respect to multiplication by a constant, and it is necessary to impose the normalization condition $\sum_\gamma K^\gamma(0) = 1$ at every iteration. This corresponds to imposing the conservation of probability. With this, it is possible to directly access steady state observables.

In practice, Eq. (23) is solved over a finite time interval. The length of this interval therefore becomes a numerical parameter with respect to which the calculation needs to be converged. However, the computational cost scales just linearly in the interval length, such that obtaining convergence is usually inexpensive compared to performing the full time propagation.

## 3.2 Adapting the noncrossing approximation to singlet weight observables

The vertex function $K_\alpha^\beta \left( t, t' \right)$ can be used to obtain the expectation value of any single-time dot operator according to Eq. (14), either exactly or within the NCA. However, nonlocal observables comprising operators from the leads—or both the dot and leads—cannot be immediately

expressed in terms of the vertex function. In this subsection, we will develop an NCA approach to the nonlocal observables needed to obtain the singlet weight: $\langle E^{\sigma\sigma'}_{\chi\chi'}\rangle$ and $\langle P^{\sigma\sigma'}_{\chi\chi'\chi''\chi'''}\rangle$. The technique is similar to that used to obtain Green's functions from the NCA [59, 60].

The main issue with the appearance of lead operators in the observable is that during the application of Wick's theorem, these can be paired with lead operators from the perturbation. Diagrammatically, this results in "hybridization lines" going from the observable at the tip of the contour to all other contour times, which, in turn, breaks up the propagators and vertex functions into smaller segments. This is schematically depicted in Fig. 2(c).

In general, an exact calculation requires that all hybridization lines between these segments and the vertex function be evaluated [78]. Because these higher-order corrections invariably involve hybridization events between propagator segments already divided by a hybridization line, they can be neglected at the NCA level. A formal derivation of this approximation proceeds by setting of $A = E^{\sigma\sigma'}_{\chi\chi'}$ or $A = P^{\sigma\sigma'}_{\chi\chi'\chi''\chi'''}$ in Eq. (12), and working out the lowest non-vanishing order of the perturbative expansion of the time-evolution operators $U(t)$ in the system–bath hybridization. The atomic propagators $g_\alpha$ are then replaced by their NCA counterparts $G_\alpha$, and propagation from the initial state is replaced by $K^\beta_\alpha$ as in Fig. 2(c). This leads to relatively straightforward, if unwieldy, expressions (with the dependence on time $t$ suppressed on the left hand side):

$$
\begin{aligned}
\langle P^{\sigma\sigma'}_{\chi\chi'\chi''\chi'''}\rangle = & -\int_0^t d\tau'_1 \int_0^{\tau'_1} d\tau'_2 \, A^{\sigma\sigma'}_{\chi\chi'\chi''\chi'''}(t,\tau'_1,\tau'_2) - \int_0^t d\tau_2 \int_0^{\tau_2} d\tau_1 \, B^{\sigma\sigma'}_{\chi\chi'\chi''\chi'''}(t,\tau_1,\tau_2) \\
& -\int_0^t d\tau_1 \int_0^t d\tau'_1 \, C^{\sigma\sigma'}_{\chi\chi'\chi''\chi'''}(t,\tau_1,\tau'_1) - \int_0^t d\tau_1 \int_0^t d\tau'_1 \, D^{\sigma\sigma'}_{\chi\chi'\chi''\chi'''}(t,\tau'_1,\tau_1),
\end{aligned}
\tag{24}
$$

and

$$
\begin{aligned}
\langle E^{\sigma\sigma'}_{\chi\chi'}\rangle = & -\int_0^t d\tau'_1 \int_0^{\tau'_1} d\tau'_2 \, G_\sigma(t-\tau'_1)\Omega^{\chi\chi'}(\tau'_1,\tau'_2)\cdot K^{\sigma'}_\alpha(t,\tau'_2) \\
& -\int_0^t d\tau_2 \int_0^{\tau_2} d\tau_1 \, G^*_{\sigma'}(t-\tau_2)\Omega^{\chi'\chi*}(\tau_2,\tau_1)\cdot K^\sigma_\alpha(\tau_1,t) \\
& +\int_0^t d\tau_1 \int_0^t d\tau'_1 \, G^*_{\sigma'}(t-\tau_1)G_\sigma(t-\tau'_1)\Theta^{\chi\chi'}_0(\tau_1,\tau'_1)K^0_\alpha(\tau_1,\tau'_1) \\
& +\int_0^t d\tau_1 \int_0^t d\tau'_1 \, G^*_{\sigma'}(t-\tau_1)G_\sigma(t-\tau'_1)\Theta^{\chi\chi'}_{\uparrow\downarrow}(\tau_1,\tau'_1)K^{\uparrow\downarrow}_\alpha(\tau_1,\tau'_1).
\end{aligned}
\tag{25}
$$

Here, the following set of auxiliary definitions has been used:

$$
\begin{aligned}
A^{\sigma\sigma'}_{\chi\chi'\chi''\chi'''} \equiv & \, G_{\sigma'}(t-\tau'_1)K^{\sigma'}_\alpha(t,\tau'_2) \\
& \times \left(\delta_{\chi''\chi'''}\bar{f}_{\chi''}\cdot\Xi^{\chi\chi'}_{\uparrow\downarrow}(\tau'_1,\tau'_2) + \delta_{\chi\chi'}f_\chi\cdot\tilde{\Xi}^{\chi'''\chi''}_0(\tau'_1,\tau'_2)\right),
\end{aligned}
\tag{26}
$$

$$
\begin{aligned}
B^{\sigma\sigma'}_{\chi\chi'\chi''\chi'''} \equiv & \, G^*_{\sigma'}(t-\tau_2)K^{\sigma'}_\alpha(\tau_1,t) \\
& \times \left(\delta_{\chi''\chi'''}\bar{f}_{\chi''}\cdot(\Xi^{\chi'\chi}_{\uparrow\downarrow})^*(\tau_2,\tau_1) + \delta_{\chi\chi'}f_\chi\cdot(\tilde{\Xi}^{\chi''\chi'''}_0)^*(\tau_2,\tau_1)\right),
\end{aligned}
\tag{27}
$$

$$
C^{\sigma\sigma'}_{\chi\chi'\chi''\chi'''} \equiv G^*_{\sigma'}(t-\tau_1)G_{\sigma'}(t-\tau'_1)K^0_\alpha(\tau_1,\tau'_1) \times \delta_{\chi\chi'}f_\chi\Theta^{\chi'''\chi''}_0(\tau_1,\tau'_1),
\tag{28}
$$

$$
D^{\sigma\sigma'}_{\chi\chi'\chi''\chi'''} \equiv G^*_{\sigma'}(t-\tau_1)G_{\sigma'}(t-\tau'_1)K^{\uparrow\downarrow}_\alpha(\tau_1,\tau'_1) \times \delta_{\chi''\chi'''}\bar{f}_{\chi''}\Theta^{\chi\chi'}_{\uparrow\downarrow}(\tau_1,\tau'_1).
\tag{29}
$$

Finally, setting $\eta_{\chi\chi'} \equiv V^*_\chi V_{\chi'}$ (with $V_\chi$ the coupling between the dot and lead orbital $\chi$), this

relies on the quantities:

$$\Xi_\alpha^{\chi\chi'}(\tau_1,\tau_2) = \eta_{\chi\chi'} \cdot f_\chi \bar{f}_{\chi'} \cdot e^{i\epsilon_\chi(t-\tau_2)} e^{-i\epsilon_{\chi'}(t-\tau_1)} \cdot G_\alpha(\tau_1-\tau_2), \tag{30}$$

$$\tilde{\Xi}_\alpha^{\chi\chi'}(\tau_1,\tau_2) = \eta_{\chi\chi'} \cdot f_\chi \bar{f}_{\chi'} \cdot e^{i\epsilon_\chi(t-\tau_1)} e^{-i\epsilon_{\chi'}(t-\tau_2)} \cdot G_\alpha(\tau_1-\tau_2), \tag{31}$$

$$\Theta_\alpha^{\chi\chi'}(\tau_1,\tau_2) = -\eta_{\chi\chi'} \cdot \Big( \delta_{\alpha,0} \cdot f_\chi f_{\chi'} \cdot e^{i\epsilon_\chi(t-\tau_2)} e^{-i\epsilon_{\chi'}(t-\tau_1)}$$
$$+ \delta_{\alpha,\uparrow\downarrow} \cdot \bar{f}_\chi \bar{f}_{\chi'} e^{i\epsilon_\chi(t-\tau_1)} e^{-i\epsilon_{\chi'}(t-\tau_2)} \Big), \tag{32}$$

$$\Omega^{\chi\chi'}(\tau_1,\tau_2) = -\Big( \Xi_{\uparrow\downarrow}^{\chi\chi'}(\tau_1,\tau_2) + \tilde{\Xi}_0^{\chi\chi'}(\tau_1,\tau_2) \Big). \tag{33}$$

We note that these expressions have relatively simple diagrammatic interpretations, which are shown in Fig. 2(c).

Given the results of this subsection, the energy-resolved and position-resolved representation of the singlet weight within the NCA approach can now be obtained. These physically motivated definitions of $\chi$ will now be discussed in Secs. 3.3 and 3.4.

## 3.3 Energy-resolved singlet weights

In the energy representation of the singlet weight, the index $\chi$ represents a single-particle energy $\epsilon$ in one of the leads, $L$ or $R$. The observable isolates all contributions to $s(\epsilon)$ from a narrow range of lead energies surrounding $\epsilon$. The quantities to be evaluated are then

$$\langle P_{L/R}^{\sigma\sigma'}\rangle(\epsilon,t) = \sum_{k\in L/R} \delta(\epsilon-\epsilon_k) \cdot \langle P_{kkkk}^{\sigma\sigma'}\rangle(t),$$
$$\langle E_{L/R}^{\sigma\sigma'}\rangle(\epsilon,t) = \sum_{k\in L/R} \delta(\epsilon-\epsilon_k) \cdot \langle E_{kk}^{\sigma\sigma'}\rangle(t). \tag{34}$$

Expressions for the expectation values appearing here within the NCA were given in Sec. 3.2. To change the generic $\chi$ indices to lead level indices $k$, it is sufficient replace $\eta_{\chi\chi'}$ by $\Gamma_{L/R}(\epsilon)\delta_{kk'}$ in Eqs. (30)–(33). Given this, it is only necessary to evaluate Eqs. (24) and (25) for $k = k' = k'' = k'''$, and the calculation scales linearly with the number of single-particle energies used to describe the lead. Using the definition in Eq. (11), it is now straightforward to write the energy-resolved singlet weight in each lead in the form

$$s_{L/R}(\epsilon,t) = \frac{1}{2}\Big( \langle P_{L/R}^{\uparrow\uparrow}\rangle(\epsilon,t) + \langle P_{L/R}^{\downarrow\downarrow}\rangle(\epsilon,t) - \langle E_{L/R}^{\uparrow\downarrow}\rangle(\epsilon,t) - \langle E_{L/R}^{\downarrow\uparrow}\rangle(\epsilon,t) \Big). \tag{35}$$

In the case of the energy-resolved singlet weight, Eqs. (30)–(33) only depend on the difference between the two time arguments, such that a steady state formulation for the energy-resolved singlet weight becomes straightforward. In the steady state, Eqs. (24) and (25) can

be rewritten as follows:

$$\langle P_{t\to\infty}^{\sigma\sigma'}\rangle(\epsilon) = -\int_0^\infty d\tau \int_0^\infty d\Delta\tau \Big\{$$
$$G_{\sigma'}(\tau)\Big(\bar{f}_{L/R}(\epsilon)\cdot\Xi_{\uparrow\downarrow}(\epsilon,\Delta\tau) + f_{L/R}(\epsilon)\cdot\tilde{\Xi}_0(\epsilon,\Delta\tau)\Big)K^{\sigma'}(\tau+\Delta\tau)$$
$$+ G_{\sigma'}^*(\tau)\Big(\bar{f}_{L/R}(\epsilon)\cdot\Xi_{\uparrow\downarrow}^*(\epsilon,\Delta\tau) + f_{L/R}(\epsilon)\cdot\tilde{\Xi}_0^*(\epsilon,\Delta\tau)\Big)K^{\sigma'}(-\tau-\Delta\tau)\Big\} \qquad (36)$$
$$-\int_0^\infty d\tau \int_{-\tau}^\infty d\Delta\tau \Big\{ G_{\sigma'}^*(\tau)G_{\sigma'}(\tau+\Delta\tau)\cdot f_{L/R}(\epsilon)\Theta_0(\epsilon,\Delta\tau)\cdot K^0(\Delta\tau)$$
$$+ G_{\sigma'}^*(\tau)G_{\sigma'}(\tau+\Delta\tau)\cdot\bar{f}_{L/R}(\epsilon)\Theta_{\uparrow\downarrow}(\epsilon,\Delta\tau)\cdot K^{\uparrow\downarrow}(\Delta\tau)\Big\},$$

$$\langle E_{t\to\infty}^{\sigma\sigma'}\rangle(\epsilon) = -\int_0^\infty d\tau \int_0^\infty d\Delta\tau \,\Big\{ G_\sigma(\tau)\cdot\Omega(\epsilon,\Delta\tau)\cdot K^{\sigma'}(\tau+\Delta\tau)$$
$$+ G_{\sigma'}^*(\tau)\cdot\Omega^*(\epsilon,\Delta\tau)\cdot K^\sigma(-\tau-\Delta\tau)\Big\}$$
$$+\int_0^\infty d\tau \int_{-\tau}^\infty d\Delta\tau \,\Big\{ G_{\sigma'}^*(\tau)G_\sigma(\tau+\Delta\tau)\cdot\Theta_0(\epsilon,\Delta\tau)K^0(\Delta\tau)$$
$$+ G_{\sigma'}^*(\tau)G_\sigma(\tau+\Delta\tau)\cdot\Theta_{\uparrow\downarrow}(\epsilon,\Delta\tau)K^{\uparrow\downarrow}(\Delta\tau)\Big\}. \qquad (37)$$

This enables the direct evaluation of the energy-resolved, steady state singlet weight in the left or right lead, $s_{L/R}(\epsilon, t\to\infty)$.

### 3.4 Position-resolved singlet weights

We now continue to the position representation of the singlet weight. The general state $\chi$ entering Eq. (6) is now identified with a local lattice orbital $x$, such that

$$s(x,t) = \frac{1}{2}\Big( \langle P^{\uparrow\uparrow}\rangle(x,t) + \langle P^{\downarrow\downarrow}\rangle(x,t) - \langle E^{\uparrow\downarrow}\rangle(x,t) - \langle E^{\downarrow\uparrow}\rangle(x,t)\Big). \qquad (38)$$

Here, the position $x$ is assumed to be specific to one of the leads, such that the subscript $L/R$ can be dropped. A position-resolved representation of the singlet weight is then encoded in the observables

$$\langle P^{\sigma\sigma'}\rangle(x,t) = \langle P_{xxxx}^{\sigma\sigma'}\rangle(t), \qquad (39)$$
$$\langle E^{\sigma\sigma'}\rangle(x,t) = \langle E_{xx}^{\sigma\sigma'}\rangle(t). \qquad (40)$$

We will now discuss their evaluation.

The expectation values $\langle P^{\sigma\sigma'}\rangle(x,t)$ and $\langle E^{\sigma\sigma'}\rangle(x,t)$ can be written in terms of the previously discussed quantities $\langle P_{kk'k''k'''}^{\sigma\sigma'}\rangle(t)$ and $\langle E_{kk'}^{\sigma\sigma'}\rangle(t)$. Consider the wavefunction $\varphi_k(x)$ associated with lead mode $k$ at position $x$. The annihilation operator in the position representation then takes the form

$$a_{x\sigma} = \sum_k \varphi_k(x)a_{k\sigma}. \qquad (41)$$

Using this transformation, the position-resolved singlet weight components in Eqs. (39) and (40) are given by

$$\langle P^{\sigma\sigma'}\rangle(x,t) = \sum_{kk'k''k'''} \varphi_k^*(x)\varphi_{k'}(x)\varphi_{k''}(x)\varphi_{k'''}^*(x)\cdot\langle P_{kk'k''k'''}^{\sigma\sigma'}\rangle(t), \qquad (42)$$
$$\langle E^{\sigma\sigma'}\rangle(x,t) = \sum_{kk'} \varphi_k^*(x)\varphi_{k'}(x)\cdot\langle E_{kk'}^{\sigma\sigma'}\rangle(t). \qquad (43)$$

Within the NCA approximation used here, only terms where at least two of the four energy indices $k$ are identical contribute to Eq. (42). The naive calculation of position-resolved singlet weights in this manner therefore scales cubically with the number of lead orbitals in the NCA, and quartically in general. It is, however, possible to perform the sums over the lead eigenstates $k$ semi-analytically before the NCA calculation, essentially carrying out the evaluation of Eqs. (39) and (40) directly in the position space. To this end, we introduce the quantities

$$\zeta_x(t-\tau) = \sum_k V_k^* \varphi_k^*(x) f_k \, e^{i\epsilon_k(t-\tau)}, \tag{44}$$

$$\xi_x(t-\tau) = \sum_k V_k \varphi_k(x) \bar{f}_k \, e^{-i\epsilon_k(t-\tau)}, \tag{45}$$

$$\Lambda_x = \sum_k |\varphi_k(x)|^2 f_k, \tag{46}$$

$$\bar{\Lambda}_x = \sum_k |\varphi_k(x)|^2 \bar{f}_k. \tag{47}$$

With their help, we can rewrite Eqs. (30)–(33) specific for the position representation, including the sum over the lead energy states, as

$$\Xi_\alpha(x, \tau_1, \tau_2) = \xi_x(t-\tau_1)\zeta_x(t-\tau_2) \cdot G_\alpha(\tau_1-\tau_2), \tag{48}$$

$$\tilde{\Xi}_\alpha(x, \tau_1, \tau_2) = \zeta_x(t-\tau_1)\xi_x(t-\tau_2) \cdot G_\alpha(\tau_1-\tau_2), \tag{49}$$

$$\Theta_\alpha(x, \tau_1, \tau_2) = -\delta_{\alpha,0} \cdot \zeta_x^*(t-\tau_1)\zeta_x(t-\tau_2) - \delta_{\alpha,\uparrow\downarrow} \cdot \xi_x^*(t-\tau_1)\xi_x(t-\tau_2), \tag{50}$$

$$\Omega(x, \tau_1, \tau_2) = -\left(\Xi_{\uparrow\downarrow}(x, \tau_1, \tau_2) + \tilde{\Xi}_0(x, \tau_1, \tau_2)\right). \tag{51}$$

Similarly, we express Eqs. (26)–(29) as

$$A_x^{\sigma\sigma'} \equiv G_{\sigma'}(t-\tau_1')K_\alpha^{\sigma'}(t, \tau_2')\left(\bar{\Lambda}_x \Xi_{\uparrow\downarrow}(x, \tau_1', \tau_2') + \Lambda_x \tilde{\Xi}_0(x, \tau_1', \tau_2')\right), \tag{52}$$

$$B_x^{\sigma\sigma'} \equiv G_{\sigma'}^*(t-\tau_2)K_\alpha^{\sigma'}(\tau_1, t)\left(\bar{\Lambda}_x \Xi_{\uparrow\downarrow}^*(x, \tau_2, \tau_1) + \Lambda_x \tilde{\Xi}_0^*(x, \tau_2, \tau_1)\right), \tag{53}$$

$$C_x^{\sigma\sigma'} \equiv G_{\sigma'}^*(t-\tau_1)G_{\sigma'}(t-\tau_1')K_\alpha^0(\tau_1, \tau_1') \times \Lambda_x \Theta_0(x, \tau_1, \tau_1'), \tag{54}$$

$$D_x^{\sigma\sigma'} \equiv G_{\sigma'}^*(t-\tau_1)G_{\sigma'}(t-\tau_1')K_\alpha^{\uparrow\downarrow}(\tau_1, \tau_1') \times \bar{\Lambda}_x \Theta_{\uparrow\downarrow}(x, \tau_1, \tau_1'). \tag{55}$$

As only quantities incorporating the sum over the different lead eigenstates contribute, it is numerically feasible to treat extended systems comprising tenths of thousands of lead sites and beyond (cf. Sec. 4). Still, evaluation in position space requires the calculation of the eigenfunctions $\varphi_k(x)$ of the nointeracting lead Hamiltonian. This is essentially a tight-binding calculation, a computational task that scales cubically with the number of lead orbitals when done numerically. For periodic leads, however, it is possible to obtain converged results directly at the thermodynamic limit, and for simple systems like the 1D chains used here analytical expressions are readily available.

Finally, we remark that it is also possible to solve directly for the steady state in position space. Here, we exploit the fact that the quantities introduced in Eqs. (48)–(51) factorize into parts that depend on $t-\tau_1$, $t-\tau_2$, and $\tau_1-\tau_2$. Using this structure, we can express the

position-resolved singlet in steady state as

$$
\begin{aligned}
\langle P_{t\to\infty}^{\sigma\sigma'}\rangle(x) = &-\int_0^\infty d\tau \int_0^\infty d\Delta\tau \Big\{ \\
& G_{\sigma'}(\tau)\big(\bar\Lambda_x \xi_x(\tau)G_{\uparrow\downarrow}(\Delta\tau)\zeta_x(\mathcal{T}) + \Lambda_x \zeta_x(\tau)G_0(\Delta\tau)\xi_x(\mathcal{T})\big)K^{\sigma'}(\mathcal{T}) \\
& + G_{\sigma'}^*(\tau)\big(\bar\Lambda_x \xi_x^*(\tau)G_{\uparrow\downarrow}^*(\Delta\tau)\zeta_x^*(-\mathcal{T}) + \Lambda_x \zeta_x^*(\tau)G_0^*(\Delta\tau)\xi_x^*(-\mathcal{T})\big)K^{\sigma'}(-\mathcal{T}) \Big\} \quad (56) \\
& + \int_0^\infty d\tau \int_{-\tau}^\infty d\Delta\tau \Big\{ G_{\sigma'}^*(\tau)\zeta_x^*(\tau)G_{\sigma'}(\mathcal{T})\zeta_x(\mathcal{T})\cdot \Lambda_x \cdot K^0(\Delta\tau) \\
& \qquad\qquad\qquad + G_{\sigma'}^*(\tau)\xi_x^*(\tau)G_{\sigma'}(\mathcal{T})\xi_x(\mathcal{T})\cdot \bar\Lambda \cdot K^{\uparrow\downarrow}(\Delta\tau) \Big\},
\end{aligned}
$$

$$
\begin{aligned}
\langle E_{t\to\infty}^{\sigma\sigma'}\rangle(x) = &\int_0^\infty d\tau \int_0^\infty d\Delta\tau \Big\{ \\
& G_\sigma(\tau)\big(\xi_x(\tau)G_{\uparrow\downarrow}(\Delta\tau)\zeta_x(\mathcal{T}) + \zeta_x(\tau)G_0(\Delta\tau)\xi_x(\mathcal{T})\big)K^{\sigma'}(\mathcal{T}) \\
& + G_{\sigma'}^*(\tau)\big(\xi_x^*(\tau)G_{\uparrow\downarrow}^*(\Delta\tau)\zeta_x^*(-\mathcal{T}) + \zeta_x^*(\tau)\cdot G_0^*(\Delta\tau)\cdot \xi_x^*(-\mathcal{T})\big)K^{\sigma}(-\mathcal{T}) \Big\} \quad (57) \\
& - \int_0^\infty d\tau \int_{-\tau}^\infty d\Delta\tau \Big\{ G_{\sigma'}^*(\tau)\zeta_x^*(\tau)G_\sigma(\mathcal{T})\zeta_x(\mathcal{T})\cdot K^0(\Delta\tau) \\
& \qquad\qquad\qquad + G_{\sigma'}^*(\tau)\xi_x^*(\tau)G_\sigma(\mathcal{T})\xi_x(\mathcal{T})\cdot K^{\uparrow\downarrow}(\Delta\tau) \Big\},
\end{aligned}
$$

where $\mathcal{T} = \tau + \Delta\tau$.

We note that in order to obtain accurate steady state results at the thermodynamic limit, the underlying microscopic model for the leads needs to be large compared to the decoherence time multiplied by the Fermi velocity. In the Kondo regime in particular, systems must be considered that are large compared to the size of the Kondo cloud. It is then possible to directly calculate the position-resolved representation of the steady state singlet weight $s(x, t \to \infty)$.

## 4 Results

In this section, we present the energy- and position-resolved dynamics and steady state of the singlet weight in the nonequilibrium Anderson impurity model within the NCA framework. To simplify the discussion, we will model the leads as two semi-infinite 1D chains (see Fig. 1). This is by no means required by either the definition of the singlet weight or the NCA formalism, though it could be advantageous within matrix product state methods. Each lead site will be assumed to comprise a single orbital with on-site energy $\epsilon_b$, coupled with strength $t_b$ to its nearest neighbors. We will assume that in each lead, $\epsilon_b$ is pinned to the chemical potential in that lead, such that the isolated lead is always half occupied. In the limit of an infinitely long chain, the coupling density can be evaluated analytically [96–98] and assumes the form

$$
\Gamma_\ell(\epsilon) = \begin{cases} \frac{\sqrt{4|t_b|^2 - (\epsilon - \mu_\ell)^2}}{4t_b}\Gamma & \text{for } |\epsilon - \mu_\ell| < 2|t_b|, \\ 0 & \text{otherwise.} \end{cases} \quad (58)
$$

The maximum coupling strength to each of the two leads is therefore $\Gamma/2$. This determines the coupling between the dot and the lead site adjacent to it, $t_0 = \sqrt{t_b \Gamma}$. We employ $\Gamma$ as our energy scale. We set $t_b = 10\Gamma$ such that the lead bandwidth is $40\Gamma$. The on-site energy at the dot is $\epsilon_D = -4\Gamma$ and the Coulomb interaction $U = 8\Gamma$, such that we are investigating the particle–hole symmetric scenario. We also apply a symmetric bias across the junction by setting $\mu_{L/R} = \pm\Phi/2$, where $\Phi$ is the bias voltage.

We use the Kondo temperature as a measure for the emergence of correlation effects. The Kondo temperature is a crossover scale and its definition carries a degree of arbitrariness. A commonly used large $U$ estimate based on the Bethe ansatz suggests a Kondo temperature $T_K \approx 0.8\Gamma$ at equilibrium [3]. We found that this is consistent with the temperature at which the Abrikosov–Suhl resonance appears in the spectral function (data not shown) and in the differential conductance. This is also consistent with the operational definition used in Ref. [99]. We stress once again that the NCA does not generally produce the exact Kondo temperature and demonstrates other failures [59, 67, 72, 94, 95]. However, it does provide a qualitative picture. A systematic validation of the results upon comparison with more refined methods is left for future work.

Equilibrium Kondo correlations are expected to be characterized by a length scale $\xi = v_f / T_k$, where $v_f$ is the Fermi velocity in the noninteracting Fermionic leads [7, 21]. If we assume that the spacing between sites is $a$, the Fermi velocity can be written as $v_f = 2t_b a$, such that at the parameters above one expects $\xi/a \approx 25$. In the finite time calculations finite leads were used. Simulation were run for several chains lengths at the timescale shown, and we found the data to be converged with $\sim 350$ sites on each side. At any finite chain lengths, reflections from the ends of the chain eventually develop (not shown); the effect of such reflections on the charge density has previously been studied [100, 101]. For position-resolved calculations in the steady state, we found that convergence requires simulating chains with $\sim 12,000$ sites. The energy-resolved singlet weights shown, on the other hand, are always calculated directly in the thermodynamic limit.

We will study the system at four representative lead temperatures, $T = 0.2\Gamma$, $0.5\Gamma$, $1.0\Gamma$, and $5.0\Gamma$. Respectively, these temperatures are well below the Kondo crossover scale; close to but still below Kondo; slightly above Kondo; and well above Kondo. In Sec. 4.1 we will consider relaxation to equilibrium from an initially factorized state. In Sec. 4.2 we will apply a voltage bias between the leads to drive the system to a nonequilibrium steady state.

## 4.1 Relaxation to equilibrium

We begin by examining the energy-resolved singlet weight $s_{L/R}(\epsilon, t)$ without a bias voltage, i.e. for $\mu_{L/R} = \Phi = 0$. This obviates the distinction between the left and right leads, and we therefore drop the corresponding index for the remainder of this section and write the observable as $s(\epsilon, t)$.

In the top panels of Fig. 3(a) and (b), the time and frequency dependence of $s(\epsilon, t)$ is plotted at $T = 0.2\Gamma$, well below the Kondo temperature. Two different initial states are shown: the dot is initially empty in (a), and fully magnetized in (b). The lower panels present cuts at constant time across the same data. Additionally, the equilibrium steady state that eventually develops at the long time limit is shown.

The most prominent feature at long times (dashed lines), where the result is independent of the initial conditions, is the central peak at $\epsilon = 0$. We associate this peak with singlet correlations driven by the Kondo effect. The dip at its center, which is more prominent at shorter times, is due to the classical correlations subtracted in Eq. (10). Due to the factor $f(\epsilon)\bar{f}(\epsilon)$ in our definition of the classical part, the width of the dip is determined by the temperature $T$. While classical correlations dominate $s(\epsilon, t)$ at short times, they eventually mostly fade away, leaving behind an almost pure peak representing non-classical singlet correlations. Notice, however, that the NCA is prone to overestimating relaxation time of the system [77].

When the dot is initially unoccupied, a large feature appears after a timescale $t \sim 1/\Gamma$ around $\epsilon = \epsilon_D$. This feature decreases with time, and does not appear at all in the initially magnetized state. This phenomenon is easy to understand. Consider the two electrons initially incoherently occupying the lead orbital with single-particle energy $\epsilon$. These electrons are of opposite spins. At short times, before interactions take effect, both electrons can resonantly

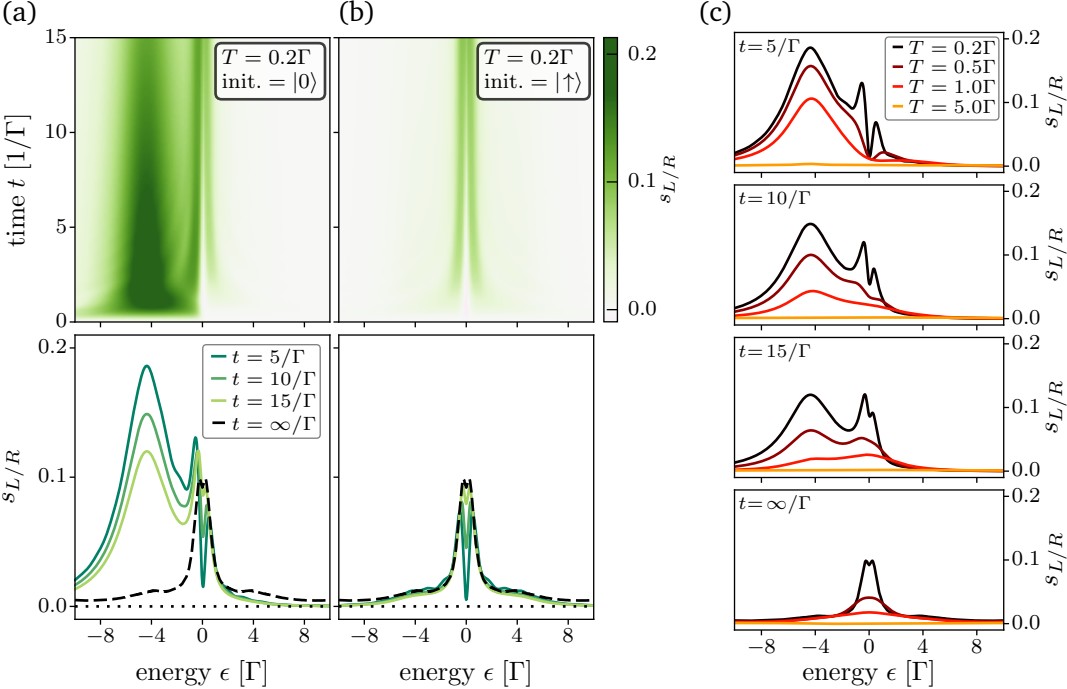

Figure 3: Formation of equilibrium ($\Phi = 0$) energy-resolved singlet weight. (a) Top: Dynamics with an initially unpopulated dot at temperature $T = 0.2\Gamma < T_K$. Bottom: Cuts across the data at several representative finite times, with the dashed black line corresponding to steady state. (b) Same as (a), but with an initially magnetized dot. (c) Singlet weight at different times and temperatures. Time increases towards lower panels, with the bottom panel at steady state.

enter the unoccupied dot. For some timescale, until decoherence takes place, these electrons can be expected to maintain their original singlet correlations while being split between the dot and lead orbitals. Preliminary investigations of the scaling behavior of this timescale with $U$ reveal a linear relationship between the relaxation time and $1/T_K$. The relaxation dynamics is therefore fully determined by the Kondo temperature, albeit with a prefactor that remains to be understood. Further analysis will await numerically exact results. When the dot eventually stabilizes in the half-occupied singlet-state, this effect is suppressed because the dot and lead share the same occupancy. From an analogous argument, it is easy to see that a corresponding transient effect must appear at $\epsilon = -\epsilon_D$ for the initially doubly-occupied state (not shown here).

An essential facet of Kondo physics is its dependence on temperature. In Fig. 3(c), we present a series of plots at different lead temperatures for the initially empty dot. These are shown at constant times, increasing towards equilibrium at progressively lower panels. Higher temperatures suppress singlet correlations in essentially all cases. While at short times, non-interacting contributions to $s(\epsilon)$ at $\epsilon \approx \epsilon_D$ form at all temperatures, the Kondo correlations at $\epsilon \approx 0$ form only below the Kondo temperature at both short and long timescales. The dip due to classical spin–spin correlations that appears at low frequencies is only visible when the temperature $T$ is substantially smaller than the Kondo temperature $T_K$, such that the dip is narrower than the Kondo peak. If the temperature was even lower, the dip would eventually become too narrow to be distinguished.

A final interesting feature should be noted. Surrounding the main Kondo peak is a weaker, wider feature extending from $\epsilon \approx -U/2$ to $\epsilon \approx U/2$. This implies that, at least within the NCA and at finite temperature, a remnant of singlet correlations exists throughout the range

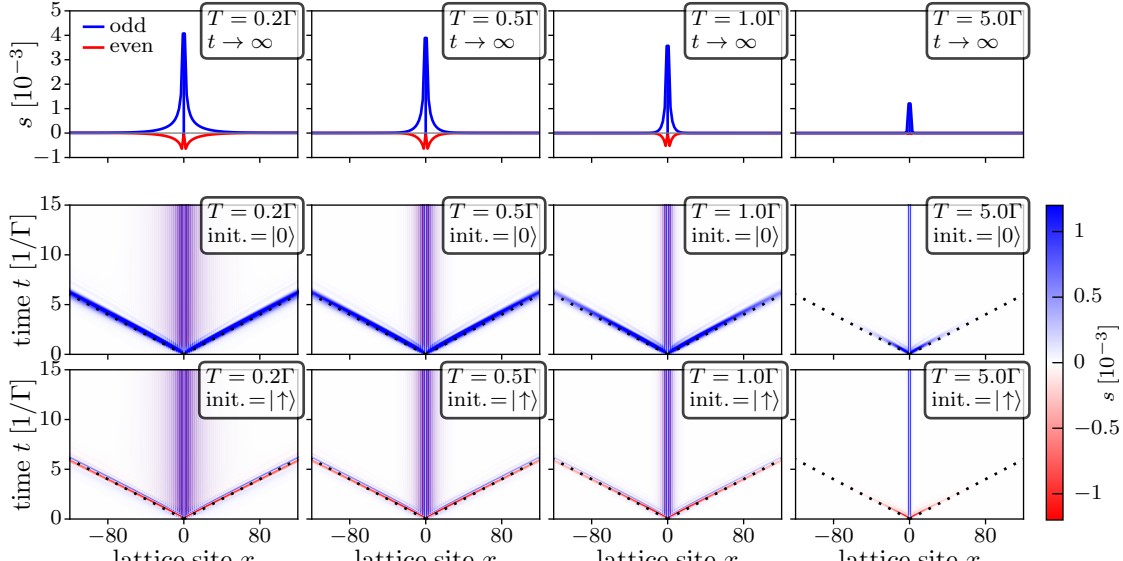

Figure 4: Formation of equilibrium ($\Phi = 0$) position-resolved singlet weight $s(x,t)$, with parameters as in Fig. 3, at two initial dot conditions: unoccupied (middle row) and magnetized (bottom row). The steady state is depicted in the top row, separated into even and odd sites. A series of temperatures is shown, increasing from left to right. The dot is located at $x = 0$. The black dotted lines in the middle and bottom row panels indicate the location $v_f t$.

of energies accessible by dot excitations, and not just within the Kondo peak. Nevertheless, the specificity with which $s(\epsilon, t \to \infty)$ corresponds to our intuitive picture of Kondo physics is striking: for example, there are no side bands, as one would observe in a spectral function. The energy-resolved singlet weight therefore remains an excellent diagnostic for the Kondo effect.

We now continue to the position-resolved singlet weight $s(x,t)$. In the 1D case considered here, we denote with $x \in \mathcal{Z}/0$ the displacement of a site in lead $L/R$ for negative/positive sign from the impurity, and use $x = 0$ to refer to the impurity site itself. Fig. 4 shows the corresponding dynamics and the steady state. We once again focus on dynamics up to time $t = 15/\Gamma$; this should be compared with Fig. 3, which shows the same time scale. The results are characterized by an even–odd structure that has previously been discussed in the literature [7, 9, 14, 27, 29, 30, 102, 103].

Perhaps the easiest feature to understand is the light cone, which appears at all parameters. It corresponds to a wavefront of singlet correlations propagating into the leads at the Fermi velocity after the coupling is activated. The magnitude of the cone structure fades with increased temperature, but, in the 1D leads discussed here, it does not rapidly decay with time and distance from the impurity. Moreover, the light cone is sensitive to the initial dot state. The wavefront's propagation obeys the Lieb–Robinson bounds, and is directly related to the spreading of spin–spin correlations that has been previously described in the literature for the single lead case [29, 30, 104–107]. Outside the light cone, one can note the formation of minor correlations due to the initial spatial entanglement within the noninteracting baths. For spin–spin correlations, this has been previously observed and analyzed [29, 30, 107]. It is interesting to note, though perhaps not particularly surprising, that the same physical picture emerges from singlet correlations.

Another conspicuous feature is the correlations that form near the impurity, at

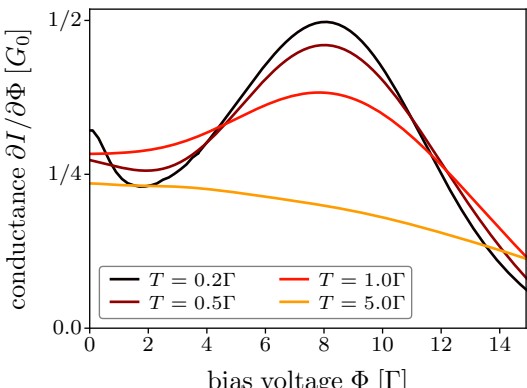

Figure 5: Steady state conductance as a function of bias voltage $\Phi$ within the NCA at a series of temperatures.

$x \lesssim 25 \approx \xi_K/a$, and which quickly establish after the light cone has reached the respective lead sites. Previous work has associated similar structures in the spin–spin correlations with the Kondo cloud [29, 30, 101, 107, 108]. Also, the dynamical generation of short-time nonequilibrium density oscillations and spin correlations after a quench has been explored, showing some evidence of cloud formation [100, 101]. The structure of the this central feature and its dependence on temperate is most easily studied based on the steady state results in the top row of Fig. 4, which are separated into the individual contributions from even and odd chain sites. We observe that the magnitude and the extent of the central feature decreases with increasing temperature, and that this feature essentially disappears at higher temperatures. In particular, for the highest temperate considered here, all that remains is a minor contribution from the terminal sites immediately adjacent to the impurity. Contributions from even chain sites are completely absent. This is remarkable, because the energy-resolved representation of the singlet weight at high temperatures is essentially zero, which indicates that the energy- and the position-resolved representations provide complementary information and are not straightforwardly mapped onto one another.

## 4.2 Nonequilibrium driving

Next, we investigate the influence of a nonzero bias voltage $\Phi$ on the singlet weight. The span of voltages we discuss corresponds to typical regimes in quantum transport scenarios, where biases ranging from linear response to Coulomb blockade can be applied to the system. To identify these relevant parameter regimes, it is useful to consider the conductance as a function of bias voltage, as shown in Fig. 5. The peak at $\Phi = 8\Gamma$ that appears at all temperatures below $T = 5.0\Gamma$ corresponds to the onset of resonant transport. When $T = 5.0\Gamma$, essentially all features are washed out by thermal broadening. At the two temperatures below Kondo, $T = 0.5\Gamma$ and more noticeably $T = 0.2\Gamma$, low bias conductance is enhanced. The enhanced conductance is due to the emergence of the Kondo resonance for temperatures below the Kondo temperature. In the low temperature limit, the Kondo resonance leads to a unitary conductance [109, 110] $G_0 = 1/2\pi$. Here, the conductance is substantially smaller because we are only at the edge of the Kondo regime, where the NCA method is still expected to be reliable. We will therefore focus most of our analysis on a low voltage within the Kondo feature, $\Phi = 1\Gamma$; an intermediate voltage in the nonresonant transport regime beyond it, $\Phi = 5\Gamma$; and a large voltage resulting in resonant transport, $\Phi = 10\Gamma$. As will be shown below, each of these regimes is characterized by a different dependence of the singlet weight on the bias voltage. For reference, we will compare all findings to the equilibrium case, $\Phi = 0$.

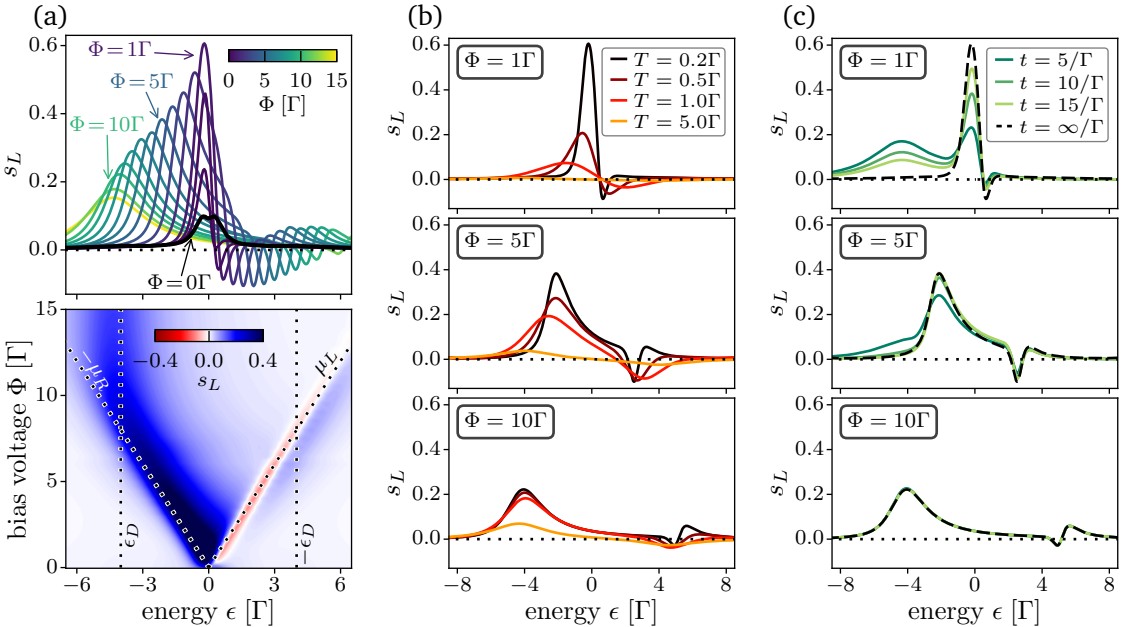

Figure 6: Nonequilibrium ($\Phi \neq 0$) energy-resolved singlet weight. (a) Steady state singlet weight of the left lead as a function of bias voltage and lead energy level for $T = 0.2\Gamma$. The top panel depicts a set of representative bias voltages, the bottom panel a comprehensive contour plot of the same data. (b) Steady state singlet weight of the left lead at a series of temperature, with bias voltage $\Phi = \Gamma$ (top panel), $\Phi = 5\Gamma$ (middle panel), and $\Phi = 10\Gamma$ (bottom panel). (c) Singlet weight in the left lead at a series of times, at temperature $T = 0.2\Gamma$ and bias voltage $\Phi = \Gamma$ (top panel), $\Phi = 5\Gamma$ (middle panel), and $\Phi = 10\Gamma$ (bottom panel). The dot is initially empty, and the dashed black lines indicate the steady state.

Fig. 6(a) presents $s_L(\epsilon)$ in the steady state at low temperature ($T = 0.2\Gamma$). The top panel shows the dependency on the bias voltage for several representative values of $\Phi$. For comparison, the equilibrium result at $\Phi = 0$ is shown in black. The bottom panel shows a contour plot of the bias dependence at the entire range of voltages. Due to particle–hole symmetry in our choice of parameters, the results for the two leads can be related to each other by the transformation $\epsilon \to -\epsilon$. The discussion can therefore be restricted to $s_L(\epsilon)$ with no loss of generality.

When a small bias voltage $\Phi \lesssim \Gamma$ in the Kondo-enhanced conductance regime is applied to the system, the symmetry of $s_L(\epsilon)$ is immediately broken. A sharp and pronounced positive peak appears at small negative frequencies, and a sharper but smaller negative peak appears at small positive frequencies. The intensity of both peaks rapidly increases with the bias voltage in this regime. Interestingly, the large positive peak corresponding to strong singlet correlation in the left lead is pinned to the chemical potential of the *right* lead, and vice versa (the bottom panel of Fig. 6(a) shows the two chemical potentials as dotted lines). This positive peak corresponds to strong, non-classical Kondo-like singlet correlations. The negative peak, which—as in equilibrium—is due to the classical spin–spin correlations, corresponds to the dip at $\epsilon = 0$ in the equilibrium curve and is pinned to the left chemical potential. Its width and location continue to essentially be determined by the lead temperature and chemical potential, as they appear within the factor $f_L \bar{f}_L$ in the classical correlation term.

The pinning of non-classical singlet correlations in each lead to the chemical potential within the other lead may be surprising, but can be justified by simple arguments. Interestingly, the mechanism for this is more closely related to the large evanescent peak at the unoccupied

initial condition in Fig. 3(a) than to the equilibrium Kondo effect. In the equilibrium case, when the dot eventually becomes singly occupied, the chemical potentials throughout the system are equalized and electrons are exchanged only as a result of undriven diffusion. In the nonequilibrium scenario, the left lead is fully occupied at the chemical potential of the right lead, while the right lead is half occupied at the same energy. The dot is half occupied at steady state, and can rapidly eject electrons into empty orbitals in the right lead through the remnants of the Kondo transmission peak. Electrons from the left lead are therefore driven to resonantly enter the dot in a process that entails transport of another electron of the same spin from the dot to the right lead, such that the dot remains singly occupied. Each such event generates a singlet between the left orbital and the dot, but no singlet between the right orbital and the dot.

The rate controlling this nonequilibrium mechanism for the formation of singlet correlations is substantially higher than that characterizing the formation of equilibrium Kondo correlations, because it is driven by the difference in occupancy between the left lead and the rest of the system at that energy, rather than just by diffusive fluctuations. This mechanism can therefore generate a large contribution to the singlet weight in the left lead at the chemical potential of the right lead, at the nonequilibrium steady state. Naturally, an analogous process in the opposite direction happens at the left lead's chemical potential, resulting in enhanced singlet correlations in the right lead.

The two singlet features remain pinned to the lead chemical potentials at larger biases $1\Gamma \lesssim \Phi \lesssim 5\Gamma$ that are still below resonant transport. However, their intensity decays with voltage and the larger peak broadens. A wider but less intense positive peak develops near the negative feature, most likely corresponding to the normal Kondo effect. For yet higher bias voltages $\Phi \gtrsim 8\Gamma$, once the resonant regime has been reached, the larger positive peak becomes pinned to the resonance energy $\epsilon_D$ and stops moving with voltage. The eventual decay and broadening of all features with increased voltage are consistent with the commonly accepted consensus that Kondo correlations cannot survive in the presence of larger voltages.

The temperature dependence of the singlet weight in the three transport regimes is explored in Fig. 6(b). In the Kondo-enhanced transport regime, the singlet weight is strongly suppressed and eventually eliminated by higher temperatures. The nonresonant transport regime at intermediate bias still shows a temperature suppression, but the nonequilibrium singlet weights, presumably associated more strongly with the nonequilibrium mechanism discussed above than with the equilibrium Kondo effect, appear to be robust at somewhat higher temperatures. In the resonant transport regime, response to small temperatures is weak, and strong suppression of singlet weights only occurs at $T = 5.0\Gamma$. This suggests (at least within the NCA) that the robustness of singlet correlations with respect to temperature may be enhanced by nonequilibrium driving.

Next, in Fig. 6(c), the quench dynamics when starting from an empty dot are presented. We observe that the relaxation timescale needed to reach the steady state is substantially decreased with increasing bias voltage. In the resonant transport regime at $\Phi = 10\Gamma$, the system already assumes its steady state by $t = 5/\Gamma$. Comparing the dynamics in Fig. 3(c) to those in Fig. 6(c) suggests that bias voltages have a substantially stronger effect on the relaxation dynamics than temperatures of similar magnitude.

The last set of results to be presented here, in Fig. 7, pertains to the spatially-resolved singlet weight $s(x, t)$. The light cone and central feature seen in subsection 4.1 are once again visible. As in the energy-resolved singlet weight, the dependence on the initial condition, which is imprinted onto the light cone in the position-resolved representation, fades more rapidly as the bias voltage is increased. The most clearly visible transient signature of nonequilibrium driving, however, is the breaking of symmetry between the left and the right lead visible for the initially unoccupied state. At short times, an initially empty dot is more

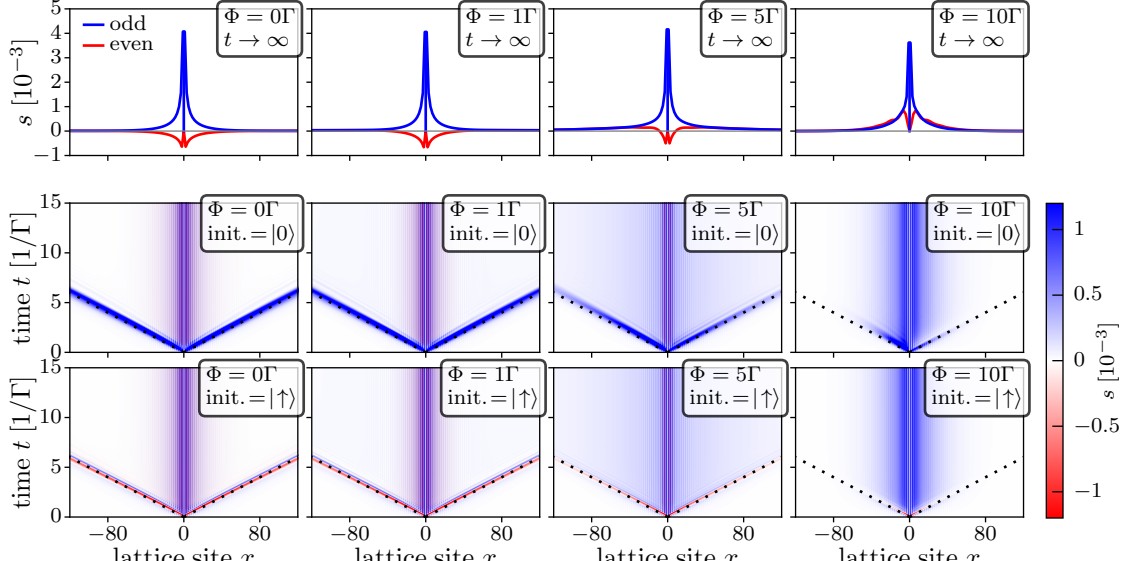

Figure 7: Formation of nonequilibrium position-resolved singlet weight $s(x,t)$, with parameters as in Fig. 6, at two initial dot conditions: unoccupied (middle row) and magnetized (bottom row). The steady state is depicted in the top row, separated into even and odd sites. A series of bias voltages is shown, increasing from left to right. The temperature is $T = 0.2\Gamma$. The dot is located at $x = 0$. The black dotted lines in the middle and bottom row panels indicate the location $v_f t$.

likely to be populated by electrons from the lead with larger chemical potential, i.e. the left lead.

The dependence of the central feature on bias voltage is most clearly visible in the steady state data in the upper panels. Here, where contributions from even and odd lead sites are shown separately, the dependence on bias voltage differs qualitatively from that on temperature (see Fig. 4). For low bias voltages, the central feature is mostly unaffected by the bias voltage. This is remarkable, since the energy-resolved singlet weight is very sensitive to a similar change in temperature within this regime. Again, this highlights the complementary information provided by the two different representations. For intermediate bias voltages still in the nonresonant transport regime, the central peak begins to extend over larger distances from the dot. This is in line with the previous argument that electrons contributing to transport form correlated singlets. This trend is eventually reversed at higher bias voltages in the resonant bias regime, presumably due to increased decoherence. For high bias voltages in the resonant transport regime, we observe that the even–odd structure breaks down, starting from higher energies. The central feature at high voltage is reminiscent of the Kondo feature at low temperatures and equilibrium, but with a different characteristic length scale, and with both even and odd sites exhibiting a positive singlet weight.

We note that the particular way in which we have applied bias voltages—by shifting the lead density of states along with the chemical potential, rather than changing the filling factor—entails that the Fermi velocity in the two noninteracting leads is unmodified. Therefore, only one correlation length is expected to remain present. This indeed appears to be the case, though a more systematic examination of this correlation length and its dependence on voltage would be interesting given a more reliable method. The question of whether multiple correlation lengths can appear when the lead filling factors are modified will be left to future work.

# 5 Conclusion

We investigated the formation of singlet correlations between electrons in an interacting Anderson impurity, and orbitals in a pair of noninteracting 1D leads coupled to it. In order to quantify this, we devised singlet weight observables comprising dot and lead degrees of freedom. Measuring these weights experimentally requires a "quantum measurement" scheme, because it contains operators that cannot be expressed as a simple correlation function. Focusing on regimes where the Kondo effect is expected to generate singlet correlations, we identified the lead orbitals that most significantly contribute to the formation of the Kondo effect, in both the energy and position representations. We presented results for the evolution of singlet weights after an impurity–lead coupling quench, and for their final steady state (or equilibrium) values in the thermodynamic limit.

In equilibrium, we showed that the energy-resolved singlet weight vanishes at high temperature, while manifesting only a single sharp peak at the chemical potential below the Kondo temperature. This allows for cleanly and precisely separating the Kondo-induced singlet correlations from other phenomena, without relying on differences between energy scales as one would in considering the spectral function. Classical correlations are also easy to quantify and remove within this scheme.

The position-resolved equilibrium singlet weight is concentrated in a well-defined region centered around the quantum dot, which forms within the light cone after a quench. The size of the region is consistent with the Kondo correlation scale, supporting the notion of a Kondo cloud. As an aim for future work, it will be of some interest to see whether sum rules for the ground state [7, 8] can be extended to finite temperatures and voltages.

We noted that corresponding equilibrium scenarios have been studied extensively in the literature using a variety of measures other than the singlet weight, often with more reliable numerical techniques. The equilibrium results presented here therefore serve only to corroborate these existing results, and do not reveal new physics. Nevertheless, they confirm that the secondary observables studied by previous authors truly correspond to quantum singlet correlations rather than classical ones.

We then investigated the effect of a nonequilibrium bias in several regimes. At low voltages, where conductance is enhanced by the bias, the steady state singlet weights are significantly enhanced compared to the equilibrium Kondo weights. Most notably, we identify an enhancement of singlet correlations in each lead, which is located at the chemical potential of the other lead. Moreover, we observe a change in the oscillatory behavior of the position resolved Kondo cloud with bias voltage. The mechanism for this is driven by transport, but is eventually overtaken by dissipation at higher voltages where resonant transport occurs.

Our work is based on a noncrossing approximation (NCA), and can therefore be expected to be only qualitatively accurate [59, 60, 67–70]. However, the model can be solved to a numerically exact level of accuracy within Inchworm Monte Carlo methods, and several other methodologies may also be applicable.

We believe this work constitutes a first step towards a deeper understanding of quantum correlation effects and their impact on impurity physics and transport, in general. This is because a wide variety of "quantum measurements" like the singlet weight can be constructed, allowing for the extraction of highly specific couplings and order parameters from either simulations or experiments. In particular, within the diagrammatic methodology used here and its numerically exact counterparts [77, 111, 112], studies of this nature are not limited to low energies, weak bias voltages or special lead geometries (like the 1D case considered here). Projective observables will therefore improve our ability to tie together intuitive insights from variational ansatzes and low energy physics to numerical simulations at finite temperatures and in nonequilibrium situations.

# Acknowledgments

The authors would like to acknowledge illuminating discussions with T. Schwartz, I. Kaminker, and Y. Sagi regarding the experimental implications of this work; and with E. Gull, M. Goldstein, L. Yankovitz, and H. Atanasova regarding theoretical aspects.

**Funding information**   A.E. was supported by the Raymond and Beverly Sackler Center for Computational Molecular and Materials Science, Tel Aviv University. G.C. acknowledges support by the Israel Science Foundation (Grants No. 1604/16 and 218/19) and by the PAZY foundation (Grant No. 308/19).

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
