# Peer review of "Resolving the nonequilibrium Kondo singlet in energy- and position-space using quantum measurements"

_SciPost Physics, doi:SciPost Phys. 10, 142 (2021)_

## Round 1 · Referee Report · Anonymous · 2021-2-9

Strengths

1) This paper contains a new twist in the investigation of the Kondo Problem, since perviously typically the spin correlation function in real space (Refs[8-10]) or the fixed point properties of the total system (Ref [5]) were addressed.
2) The authors extend their analysis to a two-lead finite bias out of equilibrium steady state situation.
3)Overall, I am impressed by the evolution of the singlet projector expectation value as function of energy. Starting at short times, correlations live in the vicinity of the single occupied excitation energy $e_D$ and evolve towards the chemical potential.
4)The authors excellently present their rather complicated calculation scheme for non-equilibrium situation, but also
use these real-time integral equation to approach the thermal steady state.
5)In non-equilibrium it is nicely demonstrated that the singlet weight is redistributed by temperature as well as
by finite bias voltage but in distinctly different ways. In the real space representation the authors demonstrate that the singlet expectation value oscillated between even and odd sites typical for a particle-hole symmetric lead,
and the major weight is located around the quantum dot.

Weaknesses

1)The NCA has a long history, developed independently by Grewe and Kuramoto in their seminal papers from 1983. Unfortunately, the authors do not really address the shortcomings of SNCA for finite U in reproducing the proper Kondo temperature. Vertex correction are required for including the Schrieffer-Wulff limit of the Anderson model, see Z Phys. B Condensed Matter 74, 439 (1989)
2) I do not fully understand how the Kondo temperature was extracted. Either I did not understand the parameter (using the definition found in Ref [3] I found $T_K\approx 0.05\Gamma$) or there is a problem with the estimated TK. The differentical conductance plotted in Fig 5, however, indicates that T>Tk in the calculations.
3) The y-axis in Fig 5 does not have any units.
4)The plots in Fig 7, however are so small that one cannot see the tails of s(x) at large values of x

Report

Much of the report has been included already in the summary above.

In this paper the author address the question of how a singlet in the Kondo problem
is distributed in real-space and in energy space at finite temperature. This is a new twist in the investigation of the Kondo Problem, since perviously typically the spin correlation function in real space (Refs[8-10]) or the fixed point properties of the total system (Ref [5]) were addressed. The authors extend their analysis to a two-lead finite bias out of equilibrium steady state situation.
For that purpose the authors introduce two composite operators products, $P$ and $E$, to construct a many-body generalization of a two-particle spin singlet projector. This operator is correlating the quantum dot spin state with a spin configuration of a lead orbital in real or energy space.

In order to calculate the expectation values of these projection operator the authors resort to
the simple-NCA (SNCA) in a real-time out-of equilibrium formulation.
The authors excellently present their rather complicated calculation scheme for non-equilibrium situation, but also
use these real-time integral equation to approach the thermal steady state.

The NCA has a very long history which is only presented from a perspective of recent papers. I do not want to criticise the history part of this methodology section to much since different people have different personal perspective onto the history of this almost 40 year approach. I just would like to recommend that the authors briefly addressing the relevant question of the influence of vertex correction onto equilibrium low energy scale. Maybe I overlooked something in their complicated equations such that vertex corrections are somewhere hidden in their formulation. Overall, the section 3 is well written and documents nicely the integral equation used for obtaining the final results.

I mentioned already, that I had a problem understanding the claim of $T_K\approx 0.8\Gamma$ in the paper . Using the definition found in Ref [3] one finds about $T_K\approx 0.05\Gamma$ for the parameters used by the authors which would indicate that that all plots are performed in the Kondo limit of the Anderson model hence above $T_K$ and not in the strong coupling regime of the model.
This observation is backed by the differential conductance depicted in Figure 5: the zero-bias conductance is very low compared to the charge fluctuation peak visible at about $V=U$, which is typical in the Kondo regime ($T\approx T_K$) of the model. In the strong coupling limit, however, $dI/dV$ should
approach the unitary limit of $2e^2/h$ which will be overshot by any (S)NCA treatment due to its violation of the Fermi liquid properties. Unfortunately, Fig 5 lacks the units on the y-axis so we cannot judge how far off the calculations are from the unitary conductance limit.

I have to conclude that the selected parameter operates the finite U NCA in the crossover regime about $T_K$ where we see already some onset of Kondo correlations but have not reached the strong coupling regime.
The authors can also check their numerics: The propagator spectral functions in equilibrium (not plotted here)
should exhibit a clear x-ray edge threshold behavior which can be fitted to a power law at $T\ll T_K$. At finite $T$ this power-law is washed out (see Mueller-Hartmann 1984 or the spectra plotted in Z Phys. B Condensed Matter 74, 439 (1989).

Overall, I am impressed by the evolution of the singlet projector expectation value as function of energy. Starting at short times, correlations live in the vicinity of the single occupied excitation energy $e_D$ and evolve towards the chemical potential. The interesting question arises whether there exist a sum rule by integrating over the spectrum $s_{L/R}(e)$ or $s_{L/R}(x)$ in a similar fashion as for the spatially resolved spin correlation function as pointed out by Affleck and others, for example Ref [9] for a singlet ground state. Deviations from such a sum rule could serve as an indicator for the deviations from the ground state by finite temperature excitation.

In non-equilibrium it is nicely demonstrated that the singlet weight is redistributed by temperature as well as
by finite bias voltage but in distinctly different ways. In the real space representation the authors demonstrate that the singlet expectation value oscillated between even and odd sites typical for a particle-hole symmetric lead,
and the major weight is located around the quantum dot. The plots in Fig 7, however are so small that one cannot see the large x tails of s: it would be useful to provide a double logarithmic plot of $s(x)$
to check for potential power law decays, change of the exponent once x exceeds the estimated Kondo cloud size
as well as whether the even values change from negative to positive value.

Overall the paper is a very solid and excellent piece of work which adds a new twisted to the investigation of correlations in the Kondo problem. I can highly recommend the paper, after some minor points listed in the requested change section are addressed.

Requested changes

1) Check Tk
2) Quote the original finite U ENCA paper Z Phys. B Condensed Matter 74, 439 (1989) which states the value of$ T_K\approx \Gamma/10$ for $U/\Gamma=6$. Hence $T_K$ should be smaller for the parameters used in this paper. My guess: $T_K\approx 0.05\Gamma$.
3) Discuss the difference between a simple NCA and the ENCA with respect to the low energy scale
4) Discuss the unitary limit of the differential conduction in the model and provide proper units in Fig 5.
5) Optional: Optional: provide a log-log plot of the data in Fig 7 top for $V=0$ for revealing a potential power-low tails
which would be characteristic for the strong coupling regime with $T\ll T_K$,
and an a change of exponents inside and outside of the Kondo cloud. At high temperature I expect an exponential decay driven by temperature.

6) Optional: I am wondering about a spatial or energy sum rule of these projector weights analog to the sum rule for spin-density correlation function (see Affleck et al or Borda 2007)

---

## Round 1 · Referee Report · Anonymous · 2021-2-27

Strengths

(1) Detailed analysis of singlet correlations in the Anderson impurity model out-of-equilibrium

Weaknesses

(1) The authors focus on an Anderson impurity model in a one-dimensional configuration for which more accurate methods are available.

Report

The authors analyze singlet correlations in an Anderson impurity model out-of-equilibrium. These nonlocal singlets are what is usually called the Kondo screening cloud.
Using the NCA, the authors calculate the time-evolution of the energy-dependent and position-dependent singlet weights for two different initial states. Furthermore, they analyze the time-evolution of the singlet correlations for a driven system, where both leads have different chemical potential. They observe that the singlet weight is enhanced in this situation due to the bias and give a physical picture.

I think the paper is interesting and well written, presenting some novel results about the Kondo effect out-of-equilibrium.
I will recommend publication after the authors have addressed the following questions and comments. However, after having read the acceptance criteria and expectations for SciPost, I have the feeling that this paper is more suited for SciPost Physics Core than SciPost Physics, particularly when comparing with references 29 and 30 in the current manuscript.

Furthermore, I have the following comments and questions:

Concerning the method:
In equation 12, why is the operator on the right-hand side time-dependent, although there is the time evolution operator. If this equation is in the interaction picture, the authors should define this.

Below equation 8, the projector P^sigma,sigma^\prime selects sigma_D sigma^\prime_chi seems to be the opposite. Should it say select sigma^\prime_ D sigma_chi ?

I think it would be helpful if the authors can give a short derivation of equation 24.

The authors use the NCA to calculate nonlocal singlet correlations. As in this case, the operator describing the singlet includes a hybridization, I think that the NCA is a severe approximation. Is there a way to justify or confirm their results?
Can the authors compare their results to existing studies about the Kondo effect out-of-equilibrium?

Concerning the results:
What sets the time scale for the relaxation to the equilibrium?
The Kondo temperature is set to 0.8 Gamma, nearly Gamma. So naively, I would have expected that this energy (or better its inverse) also sets the relaxation time.
But it seems to be much longer.

How do the correlations look for an uncorrelated quantum dot (U=0)? Does it look like the results at high temperatures?

  • validity: good
  • significance: ok
  • originality: good
  • clarity: high
  • formatting: good
  • grammar: good

Author:  André Erpenbeck  on 2021-02-28  [id 1270]

(in reply to Report 2 on 2021-02-27)
Category:
remark

Dear referee,

First, we would like to thank you for your report. We appreciate your taking the time to assess our work and will respond to your comments (and those of the other referee) in detail in the full reply. In this sense, both referees' questions regarding the reliability of the NCA and the relevant time and energy scales are very pertinent and require a more technical discussion. Yet, we like to clarify some points prior to revising the manuscript, especially concerning the assessment of originality and importance and the comparison with Refs. [29-30], pointed out in the review as the main weakness of our work.

First, we want to note that our paper introduces the energy- and position-resolved singlet weights, which have to the best of our knowledge not been considered in the literature before. Refs. [29-30] consider spin–spin correlations, which (as we argue in the manuscript) do not fully capture the physical picture. The other referee report also comments on this "new twist".

Second, as both referees noted, we are investigating a scenario where the system is driven out of equilibrium by an external bias voltage between the two leads. Refs. [29-30] consider the relaxation towards equilibrium from a given initial state. To this end, Ref. [29] applies TD-NRG to an impurity coupled to a single Wilson chain, and Ref.[30] applies DMRG to an impurity coupled to a single 1D chain. While both NRG and DMRG are more reliable than the NCA when converged, it is extremely challenging to reliably study bias-driven systems at long timescales within these methods. As such, there is no counterpart (even in term of spin–spin correlations) in Refs. [29] and [30] to the main results that we present in the Sec. 4.2 of our manuscript, where we show that systems under a nonequilibrium bias voltage exhibit substantially different behavior than that of systems relaxing to equilibrium. Our study of equilibration, covered in Sec. 4.1 of the manuscript, indeed contains no new physics with respect to Refs. [29-30], and its agreement with these benchmark results serves chiefly to corroborate the reliability of our approximation.

In light of the points raised above, we urge you to reconsider your assessment of the significance and originality of our work.

Sincerely yours,
André Erpenbeck and Guy Cohen

---

## Round 2 · Referee Report · Anonymous (Referee 2) · 2021-5-6

Report

The authors have put much effort into answering the questions of the referees.
I accept the authors' explanation that this manuscript goes beyond existing nonequilibrium studies of the Kondo effect and that the singlet-weights might be a useful tool for future theoretical and experimental studies. Thus, the manuscript fulfills the criteria of SciPost Physics.

I recommend this manuscript for publication in SciPost Physics.

---

## Round 2 · Referee Report · Anonymous (Referee 1) · 2021-5-7

Report

The authors gave a elaborate answer and addressed the questions raised by both referees.

To make the long story below short: Overall it is an very good paper, employing a difficult technique to a tough problem.
The authors did their best to answer all questions to the best of their abilities. I can recommend the paper for publication in the revised version. However, personally I would reconsidering their definition of the Kondo temperature by carefully gauging it against other useful definitions that are developed for various numerical and experimental approaches.

In more detail:

The authors gave a reasonable answer concerning their definition of the Kondo temperature. Apparently, they used the definition from the Bethe-ansatz also requering $U$ to be large! A different estimate is found in chapter 3 in Hewsons book. Here one has to bear in mind that additional corrections to these large $U$ formulas come into play since $U/\pi\Gamma$ is not very large for the parameters in the manuscript. The Kondo temperature is a crossover scale and can always be defined with some arbitrariness.
I am a bit confuse by the statement in the reply: $\Delta=2\Gamma$ as stated in the replay or
$2\times \Gamma/2 =\Gamma$ as written below Eq (58) of the manuscript?

Importantly, all parameters are stated clearly so that the reader can make up her/his own mind.

The spectrum shown in Fig R1 of the reply suggests that indeed the strongly correlated regime is addressed but the AS resonance is still very small (peak hight well below the Hubbard site peaks)
and well below the zero temperature limit predicted by the Friedel sum rule.
This is very encouraging since the NCA is operated in local moment regime in the vicinity or above $T_K$ as I suspected. One can also read off that the NCA underestimates the width of the Kondo resonance as expected for a second order approach in the hybridization strength: higher order processes contribute to the resonance as well.

Just a personal remark: It might be useful for the authors to consult PRL 81, 5226 (1998) for
an operative experimental definition of $T_K$ exploiting the universality of the zero-bias conductance. It was gauged using the results of Costi and Hewson from 1994 and works remarkable well. Employing Goldhaber-Gordon's approach immediately reveals that the choice of $T$ must be above $T_K$.

Why do I emphazise the importance of a proper identification of $T_K$? That becomes clearer when looking into the real-time dynamics which is the main focus of this paper. The other referee asked
"What sets the time scale for the relaxation to the equilibrium? The Kondo temperature is set to 0.8 Gamma, nearly Gamma."
and the authors answer "At equilibrium and in the scaling limit, we typically expect all time and energy scales to be universally determined by $T_K$".

This is only correct when focusing only on the dynamics govern by the low energy excitations of the system which excludes the charge dynamics. Also the spectral function does not only contain a Kondo resonance but also high energy features whose broadening is governed by $\Gamma$. Typically NEQ dynamics of local charges are governed by $\Gamma$, even below for $T\ll T_K$, while the spin dynamics is governed by $T_K$. The reason is obvious: local charge fluctuations are suppressed in the scaling limit. Clearly the charge susceptibility is governed by $1/\Gamma$ while the spin susceptible approaches $1/T_K$ in the scaling limit.

The authors continue in their reply " However, unlike the low-energy features, the transient peak in the energy-resolved singlet weight when starting from an empty dot clearly decays much more slowly (Fig. 3(a))"

I also noticed this slow dynamics when reading the first version of the paper and that was the reason why I instigated a discussion on $T_K$. I suspected that
the spin dynamics reported by the authors indeed governed by $T_K$. However, the authors' estimate for $T_K$ is simply to high such that this point was not recognised by the authors. The authors write in their reply
"one can extract a timescale of ∼ $25\Gamma$", I guess they mean $\tau\approx 25/\Gamma $ which would
be comparable with my estimate of $T_K$ in my first report, suggesting that $\tau\approx 1/T_K$

Question: which other low energy scale should drive the long time dynamics? I suspect that there is non!

---

## Round 2 · Author Response

The resubmission letter is included in the uploaded document (pages 1-10). The revised manuscript is contained in the uploaded document on pages 11-39. As our reply to the referees includes graphs and figures, it was not possible to provide the answer in the text box. If need be, we can easily provide the corresponding source files of our reply letter.

---

## Round 2 · List of Changes

A version of the manuscript where all changes are highlighted in color is included in the uploaded document on pages 40-68.

---

## Round 3 · Author Response

\textbf{Referee 1}
\begin{quote}
The authors gave a elaborate answer and addressed the questions raised by both referees.
To make the long story below short: Overall it is an very good paper, employing a difficult technique to a tough problem. The authors did their best to answer all questions to the best of their abilities. I can recommend the paper for publication in the revised version. However, personally I would reconsidering their definition of the Kondo temperature by carefully gauging it against other useful definitions that are developed for various numerical and experimental approaches.
\end{quote}
We are thankful for this comment. Please see more about the definition of the Kondo scale and its relationship with the relaxation time below.
\begin{quote}
In more detail:
The authors gave a reasonable answer concerning their definition of the Kondo temperature. Apparently, they used the definition from the Bethe-ansatz also requering $U$ to be large! A different estimate is found in chapter 3 in Hewsons book. Here one has to bear in mind that additional corrections to these large $U$ formulas come into play since $U/\pi\Gamma$ is not very large for the parameters in the manuscript. The Kondo temperature is a crossover scale and can always be defined with some arbitrariness.
\end{quote}
It is true that the Kondo temperature is a crossover scale and its various definitions agree only up to a constant even in the scaling regime. Our estimate for the Kondo temperature is indeed based on the Bethe-ansatz in the large $U$ limit (with some corrections for intermediate $U$), but is consistent with the appearance of the Abrikosov--Suhl resonance in the spectral function. As this is an essential and potentially confusing point, we have added this information to the revised version of the manuscript.
\begin{quote}
I am a bit confuse by the statement in the reply: $\Delta=2\Gamma$ as stated in the replay or $2\times\Gamma/2=\Gamma$ as written below Eq (58) of the manuscript?
\end{quote}
We apologize for this not being entirely clear, but there is no typographic error. First off, we want to emphasize that the quantity $\Delta$ from the reply letter does not appear in the manuscript itself. It appears only in the reply letter, where we have copied the equation directly from Hewsons's book and subsequently identified the parameters in this formula with the ones used in our manuscript. The factor of $2$ in the expression $\Delta=2\Gamma$ stems from different conventions for the definition of coupling strength. This differs slightly depending on the field and the previous work that one wants to be consistent with. It has nothing to do with the factor of $2$ arising from the fact that we consider a setup comprising two leads, which sets the sum (over leads) of the maximum coupling strength to $2\times\Gamma/2=\Gamma$. We hope that this explanation clarified the meaning of the formulas provided in the manuscript and the reply letter.
\begin{quote}
Importantly, all parameters are stated clearly so that the reader can make up her/his own mind.
\end{quote}
We agree that this is crucial.
\begin{quote}
The spectrum shown in Fig R1 of the reply suggests that indeed the strongly correlated regime is addressed but the AS resonance is still very small (peak hight well below the Hubbard site peaks) and well below the zero temperature limit predicted by the Friedel sum rule. This is very encouraging since the NCA is operated in local moment regime in the vicinity or above $T_K$ as I suspected. One can also read off that the NCA underestimates the width of the Kondo resonance as expected for a second order approach in the hybridization strength: higher order processes contribute to the resonance as well.
\end{quote}
Once again, we are in perfect agreement with the referee.
\begin{quote}
Just a personal remark: It might be useful for the authors to consult PRL 81, 5226 (1998) for an operative experimental definition of $T_K$ exploiting the universality of the zero-bias conductance. It was gauged using the results of Costi and Hewson from 1994 and works remarkable well. Employing Goldhaber-Gordon's approach immediately reveals that the choice of T must be above $T_K$.
\end{quote}
We thank the referee for this comment. The 1998 Goldhaber-Gordon paper (now reference 99 in the revised manuscript) operationally defined $T_K$ as the temperature where the zero-bias conductance is $G_0/2$; here, that would give $T_K\simeq\Gamma$, consistent with our result. The expression used there, for $\epsilon_0=-U/2$, can be written as $T_K=\frac{\sqrt{\Gamma' U}}{2} e^{-\frac{\pi U}{4 \Gamma'}}$. If we had $\Gamma'=\Gamma$, this would indeed give a much lower Kondo temperature. However, the result from Hewson is very similar to this, except for a correction term that is only important at small values of $U$. It can be written as $T_K=\sqrt{\frac{\Delta U}{2}} e^{-\frac{\pi U}{8 \Delta}+\frac{\pi\Delta}{2 U}}$. Clearly this is consistent with Goldhaber-Gordon at large $U$ if and only if $\Gamma'=2\Delta$. In turn, therefore, $\Gamma'=2\Delta=4\Gamma$ and the Kondo temperature using the formula in the Goldhaber-Gordon paper is $\sim 0.58\Gamma$: reasonable, but less accurate than the number we used, because it does not include the small-$U$ correction term. This shows that we are farther from the strong coupling regime than might be thought without carefully examining the choice of units, just as the very high Kondo temperature suggests. Once again, this choice of parameters is dictated by the desire to avoid pushing against the limits of the NCA's accuracy.
\begin{quote}
Why do I emphazise the importance of a proper identification of $T_K$? That becomes clearer when looking into the real-time dynamics which is the main focus of this paper. The other referee asked "What sets the time scale for the relaxation to the equilibrium? The Kondo temperature is set to 0.8 Gamma, nearly Gamma." and the authors answer "At equilibrium and in the scaling limit, we typically expect all time and energy scales to be universally determined by $T_K$".
This is only correct when focusing only on the dynamics govern by the low energy excitations of the system which excludes the charge dynamics. Also the spectral function does not only contain a Kondo resonance but also high energy features whose broadening is governed by $\Gamma$. Typically NEQ dynamics of local charges are governed by $\Gamma$, even below for $T \ll T_K$, while the spin dynamics is governed by $T_K$. The reason is obvious: local charge fluctuations are suppressed in the scaling limit. Clearly the charge susceptibility is governed by $1/\Gamma$ while the spin susceptible approaches $1/T_K$in the scaling limit.
\end{quote}
We thank the referee for this comment. Indeed our statement was too broadly phrased and what the referee says is true, and even seen clearly in some of our own former papers.
\begin{quote}
The authors continue in their reply " However, unlike the low-energy features, the transient peak in the energy-resolved singlet weight when starting from an empty dot clearly decays much more slowly (Fig. 3(a))"
I also noticed this slow dynamics when reading the first version of the paper and that was the reason why I instigated a discussion on $T_K$. I suspected that the spin dynamics reported by the authors indeed governed by $T_K$. However, the authors' estimate for $T_K$ is simply to high such that this point was not recognised by the authors. The authors write in their reply "one can extract a timescale of $\sim 25\Gamma$", I guess they mean $\tau\approx 25/\Gamma$ which would be comparable with my estimate of $T_K$ in my first report, suggesting that $\tau\approx 1/T_K$
Question: which other low energy scale should drive the long time dynamics? I suspect that there is non!
\end{quote}
We appreciate this very insightful comment, which pushed us to reexamine some of our assumptions. The Kondo temperature is only defined up to an observable-dependent constant. To answer whether it controls the long relaxation timescale we observed, one must therefore consider scaling behavior. With this in mind, we investigated the relationship between the low-temperature decay time $\tau$ and $T_K$ (as obtained from the Bethe ansatz formula) for several values of $U$ between $4\Gamma$ and $10\Gamma$. Our preliminary results reveal that, at least within the NCA, $1/\tau$ is essentially \emph{linear} in $T_K$ over this parameter range. So, even though the $T_K$ relevant to transport and given by the formulas above is rather large, this validates the referee's suspicion! The timescale is controlled by the Kondo temperature, but the relevant Kondo scale varies by a constant factor. We have included this information in the revised version of the manuscript and thank the reviewer for this comment. However, we chose not to include the plot or investigate the prefactor in too much detail until a more reliable method is available.
Finally, we apologize for the mix-up in the units of $\tau$. Indeed, we meant $\tau\approx 25/\Gamma$.\\
\textbf{Referee 2}
\begin{quote}
The authors have put much effort into answering the questions of the referees. I accept the authors' explanation that this manuscript goes beyond existing nonequilibrium studies of the Kondo effect and that the singlet-weights might be a useful tool for future theoretical and experimental studies. Thus, the manuscript fulfills the criteria of SciPost Physics.
I recommend this manuscript for publication in SciPost Physics.
\end{quote}
We thank the referee for this comment and for recommending publication in SciPost Physics.

---

## Round 3 · List of Changes

\begin{itemize}
\item at the top of page 14:\\
We replaced the sentence
\begin{quote}
At equilibrium, these parameters suggest a Kondo temperature $T_K\approx 0.8\Gamma$ [3].
\end{quote}
by the more detailed statement:
\begin{quote}
We use the Kondo temperature as a measure for the emergence of correlation effects. The Kondo temperature is a crossover scale and its definition carries a degree of arbitrariness. A commonly used large $U$ estimate based on the Bethe ansatz suggests a Kondo temperature $T_K\approx 0.8\Gamma$ at equilibrium [3]. We found that this is consistent with the temperature at which the Abrikosov--Suhl resonance appears in the spectral function (data not shown) and in the differential conductance. This is also consistent with the operational definition used in Ref.~[99].
\end{quote}
\item at the top of page 16:\\
We have added the following statement to the revised manuscript:
\begin{quote}
Preliminary investigations of the scaling behavior of this timescale with $U$ reveal a linear relationship between the relaxation time and $1/T_K$. The relaxation dynamics is therefore fully determined by the Kondo temperature, albeit with a prefactor that remains to be understood. Further analysis will await numerically exact results.
\end{quote}
\end{itemize}
Anonymous on 2021-05-25 [id 1465]
Editorial comment: PDF version of resubmission letter added for better readability.
Attachment:
resubmission_scipost_202102_00003.pdf

---

## Editorial Decision

published